# FiLM: Frequency improved Legendre Memory Model for Long-term Time Series Forecasting

**Tian Zhou**[*†]   **Ziqing Ma**[*]   **Xue Wang**   **Qingsong Wen**   **Liang Sun**
**Tao Yao**   **Wotao Yin**   **Rong Jin**[†]
{tian.zt,maziqing.mzq,xue.w,qingsong.wen,liang.sun}@alibaba-inc.com
{tao.yao,wotao.yin,jinrong.jr}@alibaba-inc.com

## Abstract

Recent studies have shown that deep learning models such as RNNs and Transformers have brought significant performance gains for long-term forecasting of time series because they effectively utilize historical information. We found, however, that there is still great room for improvement in how to preserve historical information in neural networks while avoiding overfitting to noise presented in the history. Addressing this allows better utilization of the capabilities of deep learning models. To this end, we design a **F**requency **i**mproved **L**egendre **M**emory model, or **FiLM**: it applies Legendre Polynomials projections to approximate historical information, uses Fourier projection to remove noise, and adds a low-rank approximation to speed up computation. Our empirical studies show that the proposed FiLM significantly improves the accuracy of state-of-the-art models in multivariate and univariate long-term forecasting by (**20.3%**, **22.6%**), respectively. We also demonstrate that the representation module developed in this work can be used as a general plug-in to improve the long-term prediction performance of other deep learning modules. Code is available at https://github.com/tianzhou2011/FiLM/.

## 1   Introduction

Long-term forecasting refers to making predictions based on the history for a long horizon in the future, as opposed to short-term forecasting. Long-term time series forecasting has many key applications in energy, weather, economics, transportation, and so on. It is more challenging than ordinary time series forecasting. Some of the challenges in long-term forecasting include long-term time dependencies, susceptibility to error propagation, complex patterns, and nonlinear dynamics. These challenges make accurate predictions generally impossible for traditional learning methods such as ARIMA. Although RNN-like deep learning methods have made breakthroughs in time series forecasting Rangapuram et al. (2018); Salinas et al. (2020), they often suffer from problems such as gradient vanishing/exploding Pascanu et al. (2013), which limits their practical performance. Following the success of Transformer Vaswani et al. (2017) in the NLP and CV communities Vaswani et al. (2017); Devlin et al. (2019); Dosovitskiy et al. (2021); Rao et al. (2021), it also shows promising performance in capturing long-term dependencies for time series forecasting Wen et al. (2022b); Zhou et al. (2022); Wu et al. (2021); Zhou et al. (2021). We provide an overview of this line of work (including deep recurrent networks and efficient Transformers) in the appendix A.

In order to achieve accurate predictions, many deep learning researchers increase the complexity of their models, hoping that they can capture critical and intricate historical information. These

---

* Equal contribution
† Corresponding authors

36th Conference on Neural Information Processing Systems (NeurIPS 2022).

methods, however, cannot achieve it. Figure 1 compares the ground truth time series of the real-world ETTm1 dataset with the predictions of the vanilla Transformer model Vaswani et al. (2017) and the LSTM model Hochreiter & Schmidhuber (1997a). It is observed that the prediction is completely off from the distribution of the ground truth. We believe that such errors come from these models miscapturing noise while attempting to preserve the true signals. We conclude that two keys in accurate forecasting are: 1) how to capture critical historical information as completely as possible, and 2) how to effectively remove the noise. Therefore, to avoid a derailed forecast, we cannot improve a model by simply making it more complex. Instead, we shall consider a robust representation of the time series Wen et al. (2022a) that can capture its important patterns without noise.

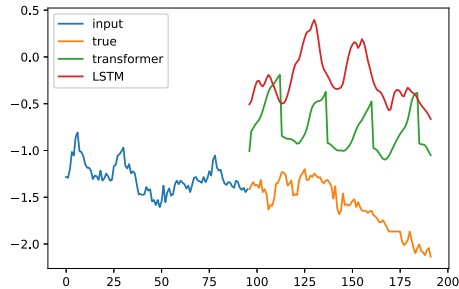 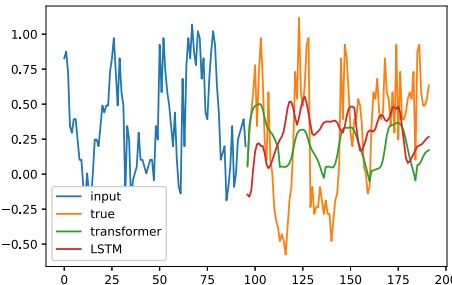

Figure 1: The discrepancy between ground truth and forecasting output from vanilla Transformer and LSTM on a real-world ETTh1 dataset Left: trend shift. Right: seasonal shift.

This observation motivates us to switch our view from long-term time series forecasting to long-term sequence compression. Recursive memory model Voelker et al. (2019); Gu et al. (2021a,b, 2020) has achieved impressive results in function approximation tasks. Voelker et al. (2019) designed a recursive memory unit (LMU) using Legendre projection, which provides a good representation for long time series. $S4$ model Gu et al. (2021a) comes up with another recursive memory design for data representation and significantly improves state-of-the-art results for Long-range forecasting benchmark (LRA) Tay et al. (2020). However, when coming to long-term time series forecasting, these approaches fall short of the Transformer-based methods' state-of-the-art performance. A careful examination reveals that these data compression methods are powerful in recovering the details of historical data compared to LSTM/Transformer models, as revealed in Figure 2. However, they are vulnerable to noisy signals as they tend to overfit all the spikes in the past, leading to limited long-term forecasting performance. It is worth noting that, Legendre Polynomials employed in LMP Voelker et al. (2019) is just a special case in the family of Orthogonal Polynomials (OPs). OPs (including Legendre, Laguerre, Chebyshev, etc.) and other orthogonal bases (Fourier and Multiwavelets) are widely studied in numerous fields and recently brought in deep learning Wen et al. (2021b); Wang et al. (2018); Gupta et al. (2021); Zhou et al. (2022); Gao et al. (2020); Gu et al. (2020). A detailed review can be found in Appendix A.

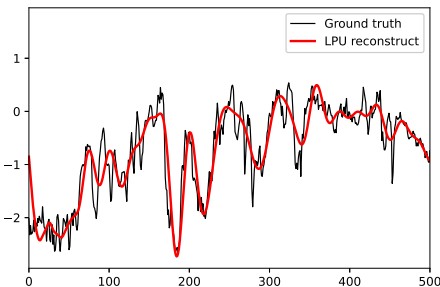 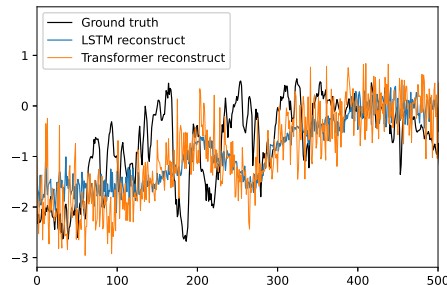

Figure 2: Data recovery with Autoencoder structure: recovery a 1024-length data with a bottleneck of 128 parameters. Left: Legendre Projection Unit. Right: LSTM and vanilla Transformer.

The above observation inspires us to develop methods for accurate and robust representations of time series data for future forecasting, especially long-term forecasting. The proposed method significantly outperforms existing long-term forecasting methods on multiple benchmark datasets by integrating those representations with powerful prediction models. As the first step towards this goal, we directly exploit the Legendre projection, which is used by LMU Voelker et al. (2019) to

update the representation of time series with fixed-size vectors dynamically. This projection layer will then be combined with different deep learning modules to boost forecasting performance. The main challenge with directly using this representation is the dilemma between information preservation and data overfitting, i.e., the larger the number of Legendre projections is, the more the historical data is preserved, but the more likely noisy data will be overfitted. Hence, as a second step, to reduce the impact of noisy signals on the Legendre projection, we introduce a layer of dimension reduction by combining Fourier analysis and low-rank matrix approximation. More specifically, we keep a large dimension representation from the Legendre projection to ensure that all the important details of historical data are preserved. We then apply a combination of Fourier analysis and low-rank approximation to keep the part of the representation related to low-frequency Fourier components and the top eigenspace to remove the impact of noises. Thus, we can not only capture the long-term time dependencies but also effectively reduce the noise in long-term forecasting. We refer to the proposed method as **F**requency **i**mproved **L**egendre **M**emory model, or **FiLM** for short, for long-term time series forecasting.

In short, we summarize the key contributions of this work as follows:

1. We propose a *Frequency improved Legendre Memory model (FiLM)* architecture with a mixture of experts for robust multiscale time series feature extraction.

2. We redesign the *Legendre Projection Unit (LPU)* and make it a general tool for data representation that any time series forecasting model can exploit to solve the historical information preserving problem.

3. We propose *Frequency Enhanced Layers (FEL)* that reduce dimensionality by combining Fourier analysis and low-rank matrix approximation to minimize the impact of noisy signals from time series and ease the overfitting problem. The effectiveness of this method is verified both theoretically and empirically.

4. We conduct extensive experiments on six benchmark datasets across multiple domains (energy, traffic, economics, weather, and disease). Our empirical studies show that the proposed model improves the performance of state-of-the-art methods by **19.2%** and **26.1%** in multivariate and univariate forecasting, respectively. In addition, our empirical studies also reveal a dramatic improvement in computational efficiency through dimensionality reduction.

## 2 Time Series Representation in Legendre-Fourier Domain

### 2.1 Legendre Projection

Given an input sequence, the function approximation problem aims to approximate the cumulative history at every time $t$. Using Legendre Polynomials projection, we can project a prolonged sequence of data onto a subspace of bounded dimension, leading to compression, or feature representation, for evolving historical data. Formally, given a smooth function $f$ observed online, we aim to maintain a fixed size compressed representation of its history $f(x)_{[t-\theta,t]}$, where $\theta$ specifies the window size. At every time point $t$, the approximation function $g^{(t)}(x)$ is defined with respect to the measure $\mu^{(t)} = \frac{1}{\theta}\mathbb{I}_{[t-\theta,t]}(x)$. In this paper, we use Legendre Polynomials of degree at most $N-1$ to build the function $g^{(t)}(x)$, i.e.

$$g^{(t)}(x) = \sum_{n=1}^{N} c_n(t) P_n\left(\frac{2(x-t)}{\theta} + 1\right),\tag{1}$$

where $P_n(\cdot)$ is the $n$-th order Legendre Polynomials. Coefficients $c_n(t)$ are captured by the following dynamic equation:

$$\frac{d}{dt}c(t) = -\frac{1}{\theta}Ac(t) + \frac{1}{\theta}Bf(t).\tag{2}$$

where the definition of $A$ and $B$ can be found in Voelker et al. (2019). Using Legendre Polynomials as a basis allows us to accurately approximate smooth functions, as indicated by the following theorem.

**Theorem 1** (*Similar to Proposition 6 in Gu et al. (2020)*)**.** *If $f(x)$ is L-Lipschitz, then $\|f_{[t-\theta,t]}(x) - g^{(t)}(x)\|_{\mu^{(t)}} \leq \mathcal{O}(\theta L/\sqrt{N})$. Moreover, if $f(x)$ has k-th order bounded derivatives, we have $\|f_{[t-\theta,t]}(x) - g^{(t)}(x)\|_{\mu^{(t)}} \leq \mathcal{O}(\theta^k N^{-k+1/2})$.*

According to Theorem 1, without any surprise, the larger the number of Legendre Polynomials basis, the more accurate the approximation will be, which unfortunately may lead to the overfitting of noisy signals in history. As shown in Section 4, directly feeding deep learning modules, such as MLP, RNN, and vanilla Attention without modification, with the above features will not yield state-of-the-art performance, primarily due to the noisy signals in history. That is why we introduce, in the following subsection, a Frequency Enhanced Layer with Fourier transforms for feature selection.

Before we close this subsection, we note that unlike Gu et al. (2021a), a fixed window size is used for function approximation and feature extraction. This is because a longer history of data may lead to a larger accumulation of noises from history. To make it precise, we consider an auto-regressive setting with random noise. Let $\{x_t\} \in \mathbb{R}^d$ be the time sequence with $x_{t+1} = Ax_t + b + \epsilon_t$ for $t = 1, 2, ...,$ where $A \in \mathbb{R}^{d \times d}$, $b \in \mathbb{R}^d$, and $\epsilon_t \in \mathbb{R}^d$ is random noise sampled from $N(0, \sigma^2 I)$. As indicated by the following theorem, given $x_t$, the noise will accumulate in $x_{t-\theta}$ over time at the rate of $\sqrt{\theta}$, where $\theta$ is the window size.

**Theorem 2.** *Let $A$ be an unitary matrix and $\epsilon_t$ be $\sigma^2$-subgaussian random noise. We have $x_t = A^\theta x_{t-\theta} + \sum_{i=1}^{\theta-1} A^i b + \mathcal{O}(\sigma\sqrt{\theta})$.*

### 2.2 Fourier Transform

Because white noise has a completely flat power spectrum, it is commonly believed that the time series data enjoys a particular spectral bias, which is not generally randomly distributed in the whole spectrum. Due to the stochastic transition environment, the real output trajectories of the forecasting task contain large volatilization and people usually only predict the mean path of them. The relatively more smooth solutions thus are preferred. According to Eq. (1), the approximation function $g^{(t)}(x)$ can be stabilized via smoothing the coefficients $c_n(t)$ in both $t$ and $n$. This observation helps us design an efficient data-driven way to tune the coefficients $c_n(t)$. As smoothing in $n$ can be simply implemented via multiplying learnable scalars to each channel, we mainly discuss smoothing $c_n(t)$ in $t$ via Fourier transformation. The spectral bias implies the spectrum of $c_n(t)$ mainly locates in the low-frequency regime and has weak signal strength in the high-frequency regime. To simplify our analysis, let us assume the Fourier coefficients of $c_n(t)$ as $a_n(t)$. Per spectral bias, we assume that there exists a $s, a_{\min} > 0$, such that for all $n$ we have $t > s, |a_n(t)| \leq a_{\min}$. An idea to sample coefficients is to keep the first $k$ dimensions and randomly sample the remaining dimensions instead of the fully random sampling strategy. We characterize the approximation quality via the following theorem:

**Theorem 3.** *Let $A \in \mathbb{R}^{d \times n}$ be the Fourier coefficients matrix of an input matrix $X \in \mathbb{R}^{d \times n}$, and $\mu(A)$, the coherence measure of matrix $A$, is $\Omega(k/n)$. We assume there exist $s$ and a positive $a_{\min}$ such that the elements in last $d - s$ columns of $A$ is smaller than $a_{\min}$. If we keep the first $s$ columns selected and randomly choose $\mathcal{O}(k^2/\epsilon^2 - s)$ columns from the remaining parts, with high probability*

$$\|A - P(A)\|_F \leq \mathcal{O}\left[(1+\epsilon)a_{\min} \cdot \sqrt{(n-s)d}\right],$$

*where $P(A)$ denotes the matrix projecting $A$ onto the column selected column space.*

Theorem 3 implies that when $a_{\min}$ is small enough, the selected space can be considered almost same as the original one.

## 3 Model Structure

### 3.1 FiLM: Frequency improved Legendre Memory Model

The overall structure of FiLM is shown in Figure 3 The FiLM maps a sequence $X \mapsto Y$, where $X, Y \in \mathbb{R}^{T \times D}$, by mainly utilizing two sub-layers: Legendre Projection Unit (LPU) layer and Fourier Enhanced Layer (FEL). In addition, to capture history information at different scales, a mixture of experts at different scales is implemented in the LPU layer. An optional add-on data normalization

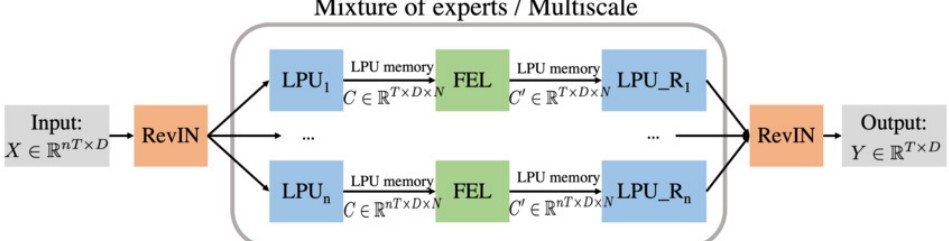

Figure 3: Overall structure of FilM (Frequency improved Legendre Memory Model). LPU: Legendre Projection Unit. LPU_R: reverse recovery of Legendre Projection. FEL: Frequency Enhanced Layer. RevIn: data normalization block. The input data is first normalized and then projected to Legendre Polynomials space (LPU memory $C$). The LPU memory $C$ is processed with FEL and generates the memory $C'$ of output. Finally, $C'$ is reconstructed and denormalized to output series with $\text{LPU}_R$. A multiscale structure is employed to process the input with length $\{T,\ 2T,\ ...\ nT\}$.

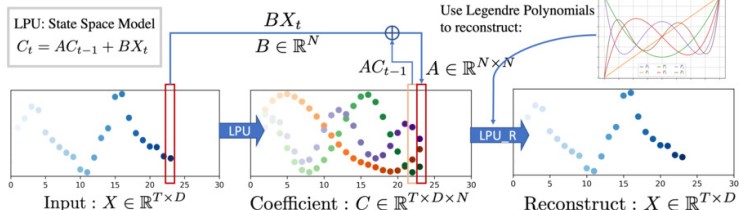

Figure 4: Structure of Legendre Projection Layer (LPU). LPU contains two states: Projection & Reconstruction. $C(t)$ is the compressed memory for historical input up to time $t$. $x(t)$ is the original input signal at time $t$. A, B are two pre-fixed projection matrices. $C(t)$ is reconstructed to original input by multiplying a discrete Legendre Polynomials matrix.

layer RevIN Kim et al. (2021) is introduced to further enhance the model's robustness. It is worth mentioning that FiLM is a simple model with only one layer of LPU and one layer of FEL.

**LPU: Legendre Projection Unit**    LPU is a state space model: $C_t = AC_{t-1} + Bx_t$, where $x_t \in \mathbb{R}$ is the input signal, $C_t \in \mathbb{R}^N$ is the memory unit, and $N$ is the number of Legendre Polynomials. LPU contains two untrainable prefixed matrices $A$ and $B$ defined as follows:

$$A_{nk} = (2n+1) \begin{cases} (-1)^{n-k} & \text{if } k \leq n \\ 1 & \text{if } k \geq n \end{cases}, B_n = (2n+1)(-1)^n. \tag{3}$$

LPU contains two stages, i.e., Projection and Reconstruction. The former stage projects the original signal to the memory unit: $C = \text{LPU}(X)$. The later stage reconstructs the signal from the memory unit: $X_{re} = \text{LPU\_R}(C)$. The whole process in which the input signal is projected/reconstructed to/from memory $C$ is shown in Figure 4.

**FEL: Frequency Enhanced Layer**

**Low-rank Approximation**    The FEL is with a single learnable weight matrix ($W \in \mathbb{R}^{M' \times N \times N}$) which is all we need to learn from the data. However, this weight could be large. We can decompose $W$ into three matrices $W_1 \in \mathbb{R}^{M' \times N' \times N'}$, $W_2 \in \mathbb{R}^{N' \times N}$ and $W_3 \in \mathbb{R}^{N' \times N}$ to perform a low-rank approximation ($N' << N$). Take Legendre Polynomials number $N = 256$ as default, our model's learnable weight can be significantly reduced to **0.4%** with $N' = 4$ with minor accuracy deterioration as shown in Section 4. The calculation mechanism is described in Figure 5.

**Mode Selection**    A subset of the frequency modes is selected after Fourier transforms to reduce the noise and boost the training speed. Our default selection policy is choosing the lowest M mode. Various selection policies are studied in the experiment section. Results show that adding some random high-frequency modes can give extra improvements in some datasets, as supported by our theoretical studies in Theorem 3.

Implementation source code for LPU and FEL are given in Appendix B.

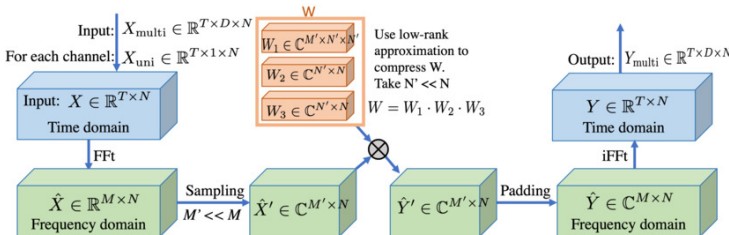

Figure 5: Structure of Frequency Enhanced Layer (FEL): Original version (use weights $W$) and Low-rank Approximation version (use weights $W = W_1 \cdot W_2 \cdot W_3$), where N is Legendre Polynomials number, M is Fourier mode number, and T is sequence length.

## 3.2 Mixture of Multiscale Experts Mechanism

The multiscale phenomenon is a unique critical data bias for time series forecasting. Since we treat history sequence points with uniform importance, our model might lack such prior. Our model implemented a simple mixture of experts strategy that utilizes the input sequence with various time horizons $\{T, 2T, ... nT\}$ to forecast the predicted horizon $T$ and merge each expert prediction with linear layer as shown in Figure 3. This mechanism improves the model's performance consistently across all datasets, as shown in Table 7.

## 3.3 Data Normalization

As Wu et al. (2021); Zhou et al. (2022) point out, time series seasonal-trend decomposition is a crucial data normalization design for long-term time series forecasting. We find that our LMU projection can inherently play a normalization role for most datasets, but lacking an explicate normalization design still hurts the robustness of performance in some cases. A simple reversible instance normalization (RevIN) Kim et al. (2021) is adapted to act as an add-on explicate data normalization block. The mean and standard deviation are computed for every instance $x_k^{(i)} \in \mathbb{R}^T$ of the input data as $\mathbb{E}_t\left[x_{kt}^{(i)}\right] = \frac{1}{T}\sum_{j=1}^{T} x_{kj}^{(i)}$ and $\text{Var}\left[x_{kt}^{(i)}\right] = \frac{1}{T}\sum_{j=1}^{T}\left(x_{kj}^{(i)} - \mathbb{E}_t\left[x_{kt}^{(i)}\right]\right)^2$. Using these statistics, we normalize the input data $x^{(i)}$ as $\hat{x}_{kt}^{(i)} = \gamma_k\left(\frac{x_{kt}^{(i)} - \mathbb{E}_t\left[x_{kt}^{(i)}\right]}{\sqrt{\text{Var}\left[x_{kt}^{(i)}\right] + \epsilon}}\right) + \beta_k$, where $\gamma, \beta \in \mathbb{R}^K$ are learnable affine parameter vectors. Then the normalized input data is sent into the model for forecasting. In the end, we denormalize the model output by applying the reciprocal of the normalization performed at the beginning.

RevIN slows down the training process by 2-5 times, and we do not observe consistent improvement in all datasets by applying RevIn. Thus, it can be considered an optional stabilizer in model training. Its detailed performance is shown in the ablation study in Table 7.

## 4 Experiments

To evaluate the proposed FiLM, we conduct extensive experiments on six popular real-world benchmark datasets for long-term forecasting, including traffic, energy, economics, weather, and disease. Since classic models such as ARIMA and simple RNN/TCN models perform inferior as shown in Zhou et al. (2021) and Wu et al. (2021), we mainly include five state-of-the-art (SOTA) Transformer-based models, i.e., FEDformer, Autoformer Wu et al. (2021), Informer Zhou et al. (2021), LogTrans Li et al. (2019), Reformer Kitaev et al. (2020), and one recent state-space model with recursive memory S4 Gu et al. (2021a), for comparison. FEDformer is selected as the main baseline as it achieves SOTA results in most settings. More details about baseline models, datasets, and implementations are described in Appendix.

### 4.1 Main Result

For better comparison, we follow the experiment settings of Informer Zhou et al. (2021) where the input length is tuned for best forecasting performance, and the prediction lengths for both training and evaluation are fixed to be 96, 192, 336, and 720, respectively.

Table 1: multivariate long-term series forecasting results on six datasets with various input length and prediction length $O \in \{96, 192, 336, 720\}$ (For ILI dataset, we set prediction length $O \in \{24, 36, 48, 60\}$). A lower MSE indicates better performance. All experiments are repeated 5 times.

| Methods | | FiLM | | FEDformer | | Autoformer | | S4 | | Informer | | LogTrans | | Reformer | |
|---|---|---|---|---|---|---|---|---|---|---|---|---|---|---|---|
| Metric | | MSE | MAE | MSE | MAE | MSE | MAE | MSE | MAE | MSE | MAE | MSE | MAE | MSE | MAE |
| ETTm2 | 96 | **0.165** | **0.256** | 0.203 | 0.287 | 0.255 | 0.339 | 0.705 | 0.690 | 0.365 | 0.453 | 0.768 | 0.642 | 0.658 | 0.619 |
| | 192 | **0.222** | **0.296** | 0.269 | 0.328 | 0.281 | 0.340 | 0.924 | 0.692 | 0.533 | 0.563 | 0.989 | 0.757 | 1.078 | 0.827 |
| | 336 | **0.277** | **0.333** | 0.325 | 0.366 | 0.339 | 0.372 | 1.364 | 0.877 | 1.363 | 0.887 | 1.334 | 0.872 | 1.549 | 0.972 |
| | 720 | **0.371** | **0.389** | 0.421 | 0.415 | 0.422 | 0.419 | 0.877 | 1.074 | 3.379 | 1.338 | 3.048 | 1.328 | 2.631 | 1.242 |
| Electricity | 96 | **0.154** | **0.267** | 0.183 | 0.297 | 0.201 | 0.317 | 0.304 | 0.405 | 0.274 | 0.368 | 0.258 | 0.357 | 0.312 | 0.402 |
| | 192 | **0.164** | **0.258** | 0.195 | 0.308 | 0.222 | 0.334 | 0.313 | 0.413 | 0.296 | 0.386 | 0.266 | 0.368 | 0.348 | 0.433 |
| | 336 | **0.188** | **0.283** | 0.212 | 0.313 | 0.231 | 0.338 | 0.290 | 0.381 | 0.300 | 0.394 | 0.280 | 0.380 | 0.350 | 0.433 |
| | 720 | 0.236 | **0.332** | **0.231** | 0.343 | 0.254 | 0.361 | 0.262 | 0.344 | 0.373 | 0.439 | 0.283 | 0.376 | 0.340 | 0.420 |
| Exchange | 96 | **0.086** | **0.204** | 0.139 | 0.276 | 0.197 | 0.323 | 1.292 | 0.849 | 0.847 | 0.752 | 0.968 | 0.812 | 1.065 | 0.829 |
| | 192 | **0.188** | **0.292** | 0.256 | 0.369 | 0.300 | 0.369 | 1.631 | 0.968 | 1.204 | 0.895 | 1.040 | 0.851 | 1.188 | 0.906 |
| | 336 | **0.356** | **0.433** | 0.426 | 0.464 | 0.509 | 0.524 | 2.225 | 1.145 | 1.672 | 1.036 | 1.659 | 1.081 | 1.357 | 0.976 |
| | 720 | **0.727** | **0.669** | 1.090 | 0.800 | 1.447 | 0.941 | 2.521 | 1.245 | 2.478 | 1.310 | 1.941 | 1.127 | 1.510 | 1.016 |
| Traffic | 96 | **0.416** | **0.294** | 0.562 | 0.349 | 0.613 | 0.388 | 0.824 | 0.514 | 0.719 | 0.391 | 0.684 | 0.384 | 0.732 | 0.423 |
| | 192 | **0.408** | **0.288** | 0.562 | 0.346 | 0.616 | 0.382 | 1.106 | 0.672 | 0.696 | 0.379 | 0.685 | 0.390 | 0.733 | 0.420 |
| | 336 | **0.425** | **0.298** | 0.570 | 0.323 | 0.622 | 0.337 | 1.084 | 0.627 | 0.777 | 0.420 | 0.733 | 0.408 | 0.742 | 0.420 |
| | 720 | **0.520** | **0.353** | 0.596 | 0.368 | 0.660 | 0.408 | 1.536 | 0.845 | 0.864 | 0.472 | 0.717 | 0.396 | 0.755 | 0.423 |
| Weather | 96 | **0.199** | **0.262** | 0.217 | 0.296 | 0.266 | 0.336 | 0.406 | 0.444 | 0.300 | 0.384 | 0.458 | 0.490 | 0.689 | 0.596 |
| | 192 | **0.228** | **0.288** | 0.276 | 0.336 | 0.307 | 0.367 | 0.525 | 0.527 | 0.598 | 0.544 | 0.658 | 0.589 | 0.752 | 0.638 |
| | 336 | **0.267** | **0.323** | 0.339 | 0.380 | 0.359 | 0.395 | 0.531 | 0.539 | 0.578 | 0.523 | 0.797 | 0.652 | 0.639 | 0.596 |
| | 720 | **0.319** | **0.361** | 0.403 | 0.428 | 0.578 | 0.578 | 0.419 | 0.428 | 1.059 | 0.741 | 0.869 | 0.675 | 1.130 | 0.792 |
| ILI | 24 | **1.970** | **0.875** | 2.203 | 0.963 | 3.483 | 1.287 | 4.631 | 1.484 | 5.764 | 1.677 | 4.480 | 1.444 | 4.400 | 1.382 |
| | 36 | **1.982** | **0.859** | 2.272 | 0.976 | 3.103 | 1.148 | 4.123 | 1.348 | 4.755 | 1.467 | 4.799 | 1.467 | 4.783 | 1.448 |
| | 48 | **1.868** | **0.896** | 2.209 | 0.981 | 2.669 | 1.085 | 4.066 | 1.36 | 4.763 | 1.469 | 4.800 | 1.468 | 4.832 | 1.465 |
| | 60 | **2.057** | **0.929** | 2.545 | 1.061 | 2.770 | 1.125 | 4.278 | 1.41 | 5.264 | 1.564 | 5.278 | 1.560 | 4.882 | 1.483 |

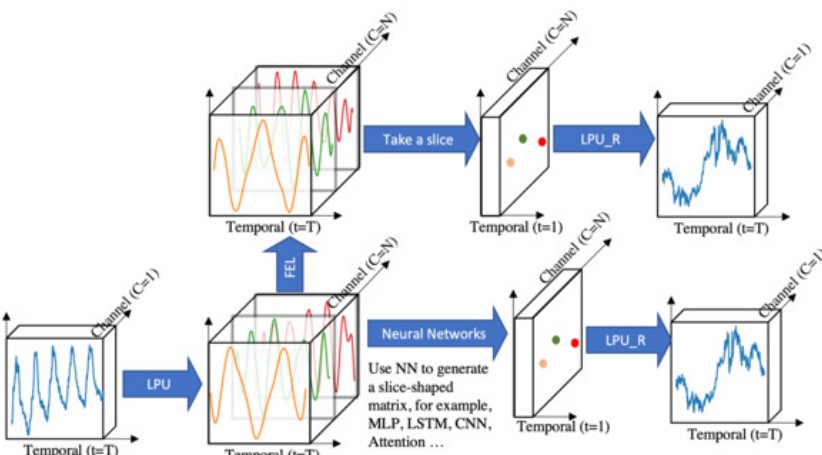

Figure 6: LPU boosting effect. LPU can serve as a plug-in block in various backbones, e.g., FEL, MLP, LSTM, CNN, and Attention. Replacing LPU with a comparable-sized linear layer will always lead to degraded performance.

**Multivariate Results**   In multivariate forecasting tasks, FiLM achieves the best performance on all six benchmark datasets at all horizons, as shown in Table 17. Compared with SOTA work (FEDformer), our proposed FiLM yields an overall **20.3%** relative MSE reduction. It is worth noting that the improvement is even more significant on some of the datasets, such as Exchange (over 30%). The Exchange dataset does not exhibit apparent periodicity, but FiLM still achieves superior performance. The improvements made by FiLM are consistent with varying horizons, demonstrating its strength in long-term forecasting. More results on the ETT full benchmark are provided in Appendix C.5.

**Univariate Results**   The benchmark results for univariate time series forecasting are summarized in Appendix C.4, Table 10. Compared with SOTA work (FEDformer), FiLM yields an overall **22.6%** relative MSE reduction. And on some datasets, such as Weather and Electricity, the improvement can reach more than 40%. It again proves the effectiveness of FiLM in long-term forecasting.

**LPU Boosting Results**   A set of experiments are conducted to measure the boosting effect of LPU when combined with various common deep learning modules (MLP, LSTM, CNN, and Attention), as shown in Figure 6. In the experiment, we compare the LPU and the comparable-sized linear layer. It is worth mentioning that LPU does not contain any learnable parameter. The results are shown in

Table 2: Boosting effect of LPU layer for common deep learning backbones: MLP, LSTM, CNN and Attention.'+' indicates degraded performance.

| Methods | | FEL | | MLP | | LSTM | | lagged-LSTM | | CNN | | Attention | |
|---|---|---|---|---|---|---|---|---|---|---|---|---|---|---|
| Compare | | LPU | Linear | LPU | Linear | LPU | Linear | LPU | Linear | LPU | Linear | LPU | Linear |
| ETTm1 | 96 | **0.030** | +38% | 0.034 | +8.0% | 0.049 | +73% | 0.093 | -21% | 0.116 | -50% | 0.243 | -81% |
| | 192 | **0.047** | +9.5% | 0.049 | +30% | 0.174 | +32% | 0.331 | -48% | 0.101 | +20% | 0.387 | -86% |
| | 336 | **0.063** | +5.8% | 0.061 | +64% | 0.119 | +84% | 0.214 | -19% | 0.122 | +25% | 1.652 | +12% |
| | 720 | **0.081** | +1.4% | 0.082 | +62% | 0.184 | +32% | 0.303 | -6.5% | 0.108 | +13% | 4.782 | -61% |
| Electricity | 96 | **0.213** | +136% | 0.431 | +121% | 0.291 | +55.6% | 0.739 | -33% | 0.310 | +43% | 0.805 | +23% |
| | 192 | **0.268** | +32% | 0.291 | +239% | 0.353 | +17% | 0.535 | +15% | 0.380 | +12% | 0.938 | +14% |
| | 336 | **0.307** | +0.1% | 0.296 | +235% | 0.436 | -6.7% | 0.517 | +23% | 0.359 | +29% | 2.043 | -54% |
| | 720 | **0.321** | +37% | 0.339 | +196% | 0.636 | -11% | 0.492 | +28% | 0.424 | +18% | 9.115 | +298% |

Table 19. For all the modules, LPU significantly improves their average performance for long-term forecasting: MLP: **119.4%**, LSTM: **97.0%**, CNN: **13.8%**, Attention: **8.2%**. Vanilla Attention has a relatively poor performance when combining the LPU, which is worth further digging.

Table 3: Low-rank Approximation (LRA) study for Frequency Enhanced Layer (FEL): Comp. K=0 means default version without LRA, 1 means the largest compression using K=1.

| Comp. K | | 0 | | 16 | | 4 | | 1 | |
|---|---|---|---|---|---|---|---|---|---|
| Metric | | MSE | MAE | MSE | MAE | MSE | MAE | MSE | MAE |
| ETTh1 | 96 | **0.371** | **0.394** | 0.371 | 0.396 | 0.371 | 0.398 | 0.400 | 0.421 |
| | 192 | 0.414 | **0.423** | 0.411 | 0.423 | 0.414 | 0.426 | 0.435 | 0.444 |
| | 336 | **0.442** | 0.445 | 0.443 | 0.446 | 0.443 | 0.444 | 0.492 | 0.478 |
| | 720 | **0.454** | **0.451** | 0.464 | 0.474 | 0.468 | 0.478 | 0.501 | 0.499 |
| Weather | 96 | **0.199** | **0.262** | 0.199 | 0.263 | 0.197 | 0.262 | 0.198 | 0.263 |
| | 192 | 0.228 | 0.288 | 0.225 | 0.285 | 0.226 | 0.285 | 0.225 | 0.286 |
| | 336 | 0.267 | 0.323 | 0.266 | 0.321 | 0.263 | 0.314 | 0.264 | 0.316 |
| | 720 | 0.319 | 0.361 | 0.314 | 0.355 | 0.315 | 0.354 | 0.318 | 0.357 |
| Parameter size | | 100% | | 1.95% | | 0.41% | | 0.10% | |

**Low-rank Approximation for FEL**    Low-rank approximation of learnable matrix in Frequency Enhanced Layer can significantly reduce our parameter size to **0.1%~0.4%** with minor accuracy deterioration. The experiment details are shown in Table 13. Compared to Transformer-based baselines, FiLM enjoys **80%** learnable parameter reduction and **50%** memory usage reductions, as shown in Appendix I, J.

Table 4: Mode selection policy study for frequency enhanced layer. Lowest: select the lowest $m$ frequency mode; Random: select $m$ random frequency mode; Low random: select the $0.8 * m$ lowest frequency mode and $0.2 * m$ random high frequency mode.

| Policy | | Lowest | | Random | | Low random | |
|---|---|---|---|---|---|---|---|
| Metric | | MSE | MAE | MSE | MAE | MSE | MAE |
| Exchange | 96 | **0.086** | **0.204** | 0.086 | 0.208 | 0.087 | 0.210 |
| | 192 | 0.188 | **0.292** | **0.187** | 0.318 | 0.207 | 0.340 |
| | 336 | 0.356 | **0.433** | 0.358 | 0.437 | **0.353** | 0.461 |
| | 720 | **0.727** | **0.669** | 0.788 | 0.680 | 0.748 | 0.674 |
| Weather | 96 | 0.199 | 0.262 | 0.197 | 0.256 | **0.196** | **0.254** |
| | 192 | **0.228** | **0.288** | 0.234 | 0.300 | 0.234 | 0.301 |
| | 336 | 0.267 | 0.323 | 0.266 | 0.319 | **0.263** | **0.316** |
| | 720 | 0.319 | 0.361 | 0.317 | 0.356 | **0.316** | **0.354** |

**Mode selection policy for FEL**    Frequency mode selection policy is studied in Table 4. The *Lowest* mode selection method shows the most robust performance. The results in *Low random* column show that randomly adding some high-frequency signal gives extra improvements in some datasets, as our theoretical studies in Theorem 3 support.

## 4.2    Ablation Study

This subsection presents the ablation study of the two major blocks (FEL & LPU) employed, the multiscale mechanism, and data normalization (RevIN).

Table 5: Ablation studies of LPU layer. The original LPU block (whose projection and reconstruction matrix are fixed) is replaced with 6 variants (**Fixed** means the matrix are not trainable. **Trainable** means the matrix is initialized with original parameters and trainable. **Random Init** means the matrix is initialized randomly and trainable). The experiments are performed on ETTm1 and Electricity with different input lengths. The metric of variants is presented in relative value ('+' indicates degraded performance, '-' indicates improved performance).

| Index | | Original | | Variant 1 | | Variant 2 | | Variant 3 | | Variant 4 | | Variant 5 | Variant 6 | |
|---|---|---|---|---|---|---|---|---|---|---|---|---|---|---|
| Projection | | Fixed | | | | Trainable | | | | | | Random Init | Linear | |
| Reconstruction | | Fixed | | Trainable | | Fixed | | Trainable | | Random Init | | – | Linear | |
| Metric | | MSE | MAE | MSE | MAE | MSE | MAE | MSE | MAE | MSE | MAE | – | MSE | MAE |
| ETTm1 | 96 | **0.030** | **0.128** | +6.0% | +3.1% | +4.6% | +3.1% | +4.7% | +2.7% | +0.7% | +0.6% | NaN | +38% | +22% |
| | 192 | **0.047** | **0.160** | -2.6% | -1.3% | +2.5% | +1.8% | -0.4% | +0.4% | -3.0% | -1.0% | NaN | +9.5% | +8.7% |
| | 336 | **0.063** | **0.185** | +4.2% | +1.8% | -2.4% | -0.4% | -6.3% | -2.5% | +2.2% | +1.4% | NaN | +5.8% | +5.0% |
| | 720 | **0.081** | **0.215** | -0.4% | -0.6% | +24% | +12% | +11% | +5.9% | NaN | NaN | NaN | +1.4% | +2.2% |
| Electricity | 96 | **0.213** | **0.328** | +15% | +6.8% | +11% | +5.2% | +11% | +5.1% | +11% | +4.8% | NaN | +136% | +58% |
| | 192 | **0.268** | **0.373** | -4.6% | -3.8% | +7.8% | +3.9% | +6.8% | +3.5% | -5.3% | -3.9% | NaN | +32% | +16% |
| | 336 | **0.307** | **0.417** | -4.2% | -5.0% | -3.9% | -6.0% | -7.2% | -8.0% | -8.5% | -9.0% | NaN | +0.1% | -5.0% |
| | 720 | **0.321** | **0.423** | +2.9% | +2.8% | +10% | +6.8% | +3.0% | +1.7% | +207% | +85% | NaN | 37% | 22% |

**Ablation of LPU**   To prove the effectiveness of LPU layer, in Table 5, we compare the original LPU layer with six variants. The LPU layer consists of two sets of matrices (Projection matrices & Reconstruction matrices). Each has three variants: Fixed, Trainable, and Random Init. In Variant 6, we use comparable-sized linear layers to replace the LPU layer. We observe that Variant 6 leads to a 32.5% degradation on average, which confirms the effectiveness of Legendre projection. The projection matrix in LPU is recursively called $N$ times ($N$ is the input length). So if the projection matrix is randomly initialized (Variant 5), the output will face the problem of exponential explosion. If the projection matrix is trainable (Variants 2, 3, 4), the model suffers from the exponential explosion as well and thus needs to be trained with a small learning rate, which leads to a plodding training speed and requires more epochs for convergence. Thus, the trainable projection version is not recommended, considering the trade-off between speed and performance. The variant with trainable reconstruction matrix (Variant 1) has comparable performance and less difficulty in convergence.

Table 6: Ablation studies of FEL layer. The FEL layer is replaced with 4 different variants: MLP, LSTM, CNN, and Transformer. S4 is also introduced as a variant. The experiments are performed on ETTm1 and Electricity. The metric of variants is presented in relative value ('+' indicates degraded performance, '-' indicates improved performance).

| Methods | | FilM | | LPU+MLP | | LPU+LSTM | | LPU+CNN | | LPU+attention | | S4 | |
|---|---|---|---|---|---|---|---|---|---|---|---|---|---|---|
| Metric | | MSE | MAE | MSE | MAE | MSE | MAE | MSE | MAE | MSE | MAE | MSE | MAE |
| ETTm1 | 96 | **0.029** | **0.127** | +0.0% | +0.0% | +12.1% | +7.1% | +13.5% | +9.5% | +1.7% | +1.6% | – | – |
| | 192 | **0.041** | **0.153** | -1.5% | -0.6% | +12.2% | +8.5% | +10.8% | +7.8% | +2.0% | +3.3% | – | – |
| | 336 | **0.053** | **0.175** | -1.7% | -1.7% | +4.5% | +4.0% | +10.8% | +6.3% | +4.5% | +2.9% | – | – |
| | 720 | **0.071** | **0.205** | -0.9% | -1.0% | +5.5% | +3.4% | +8.6% | +4.9% | +13.8% | +7.8% | – | – |
| Electricity | 96 | **0.154** | **0.247** | +155% | +81% | +160% | +84% | +330% | +155% | +242% | +119% | 128% | 83% |
| | 192 | **0.166** | **0.258** | +59% | +39% | +121% | +67% | +224% | +117% | +264% | +131% | 124% | 76% |
| | 336 | **0.188** | **0.283** | +55% | +35% | +150% | +74% | +128% | +71% | +183% | +95% | 117% | 69% |
| | 720 | **0.249** | **0.341** | +33% | +25% | +154% | +73% | +192% | +95% | +312% | +138% | 89% | 51% |

**Ablation of FEL**   To prove the effectiveness of the FEL, we replace the FEL with multiple variants (MLP, LSTM, CNN, and Transformer). S4 is also introduced as a variant since it has a version with Legendre projection memory. The experimental results are summarized in Table 6. FEL achieves the best performance compared with its LSTN, CNN, and Attention counterpart. MLP has comparable performance when the input length is 192, 336, and 720. However, MLP suffers from insupportable memory usage, which is $N^2 L$ (while FEL is $N^2$). S4 achieves similar results as our LPU+MLP variant. Among all, the LPU+CNN variant shows the poorest performance.

Table 7: Ablation studies of Normalization and Multiscale. Multiscale use 3 branches with: $T$, $2T$, and $4T$ as input sequence length. T is the predicted sequence length

| Dataset | | ETTm2 | | Electricity | | Exchange | | Traffic | | Weather | | Illness | |
|---|---|---|---|---|---|---|---|---|---|---|---|---|---|---|
| Metric | | MSE | Relative | MSE | Relative | MSE | Relative | MSE | Relative | MSE | Relative | MSE | Relative |
| Methods | Original | 0.301 | +16% | 0.195 | +4.8% | 0.534 | + 60% | 0.528 | +19% | 0.264 | +4.9% | 3.55 | + 80% |
| | Normalization | 0.268 | +3.6% | 0.199 | +6.4% | 0.456 | +36% | 0.656 | +48% | 0.256 | +1.5% | 3.36 | +70% |
| | Multiscale | 0.259 | + 0.19% | 0.187 | +0% | 0.335 | 0% | 0.541 | +22% | 0.253 | +0.50% | 2.41 | +22% |
| | With both | 0.271 | +4.7% | 0.189 | +1.5% | 0.398 | +19% | 0.442 | +0% | 0.253 | +0.4% | 1.97 | +0% |

**Ablation of Multiscale and Data Normalization (RevIN)**   The multiscale module leads to significant improvement on all datasets consistently. However, the data normalization achieves mixed performance, leading to improved performance on Traffic and Illness but a slight improvement on the rest. The Ablation study of RevIN data normalization and the mixture of multiscale experts employed is shown in Table 7.

### 4.3   Other Studies

Other supplemental studies are provided in Appendix. 1. A parameter sensitivity experiment (in Appendix F) is carried out to discuss the choice of hyperparameter for M and N, where M is the frequency mode number and N is the Legendre Polynomials number. 2. A noise injection experiment (in Appendix G) is conducted to show the robustness of our model. 3. A Kolmogorov-Smirnov (KS) test (in Appendix H) is conducted to discuss the similarity between the output distribution and the input. Our proposed FiLM has the best results on KS test, which supports our motivation in model design. 4. At last, (in Appendix J), though failing short in training speed compared to some MLP based models like N-HiTS Challu et al. (2022), FiLM is an efficient model with shorter per-step training time and smaller memory usage compared to baseline models in univariate forecasting tasks. However, it is worth mentioning that the training time will considerably prolong for multivariate forecasting tasks with many target variables.

## 5   Discussions and Conclusion

In long-term forecasting, the critical challenge is the trade-off between historical information preservation and noise reduction for accurate and robust forecasting. To address this challenge, we propose a Frequency improved Legendre Memory model, or **FiLM**, to preserve historical information accurately and remove noisy signals. Moreover, we theoretically and empirically prove the effectiveness of the Legendre and Fourier projection employed in our model. Extensive experiments show that the proposed model achieves SOTA accuracy by a significant margin on six benchmark datasets. In particular, we would like to point out that our proposed framework is rather general and can be used as the building block for long-term forecasting in future research. It can be modified for different scenarios. For example, the Legendre Projection Unit can be replaced with other orthogonal functions such as Fourier, Wavelets, Laguerre Polynomials, Chebyshev Polynomials, etc. In addition, based on the properties of noises, Fourier Enhanced Layer is proved to be one of the best candidates in the framework. We plan to investigate more variants of this framework in the future.

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
