# A Related Work

In this section, we will give an overview of the related literature in time series forecasting.

**Traditional Time Series Models**   The first generation of well-discussed time series model is the autoregressive family. ARIMA Box & Jenkins (1968); Box & Pierce (1970) follows the Markov process and build recursive sequential forecasting. However, a plain autoregressive process has difficulty in dealing non-stationary sequences. Thus, ARIMA employed a pre-process iteration by differencing, which transforms the series to stationary. Still, ARIMA and related models have the linear assumption in the autoregressive process, which limits their usage in complex forecasting tasks.

**Deep Neural Network in Forecasting**   With the bloom of deep neural networks, recurrent neural networks (RNNs) were designed for tasks involving sequential data. However, canonical RNN tends to suffer from gradient vanishing/exploding problems with long input because of its recurrent structure. Among the family of RNNs, LSTM Hochreiter & Schmidhuber (1997b) and GRU Chung et al. (2014) proposed gated structure to control the information flow to deal with the gradient vanishing/exploration problem. Although recurrent networks enjoy fast inference, they are slow to train and not parallelizable. Temporal convolutional network (TCN) Sen et al. (2019) is another family for sequential tasks. However, limited to the reception field of the kernel, the long-term dependencies are hard to capture. Convolution is a parallelizable operation but expensive in inference.

**Transformers**   With the innovation of Transformers in natural language processing Vaswani et al. (2017); Devlin et al. (2019) and in computer vision tasks Dosovitskiy et al. (2021); Rao et al. (2021), they are also discussed, renovated, and applied in time series forecasting Wen et al. (2022b), especially the main attention module. Some works use temporal attention Qin et al. (2017) to capture long-range dependencies for time series. Others use the backbone of Transformer. Transformers usually employ an encoder-decoder architecture, with the self-attention and cross-attention mechanisms serving as the core layers. Li et al. (2019) invents a logsparse attention module to deal with the memory bottleneck for Transformer model. Kitaev et al. (2020) uses locality-sensitive hashing to replace the attention module for less time complexity. Zhou et al. (2021) proposes a probability sparse attention mechanism to deal with long-term forecasting. Wu et al. (2021) designs a decomposition Transformer architecture with an Auto-Correlation mechanism as an alternative for the attention module. Liu et al. (2022) designs a low-complexity Pyramidal Attention for the long-term time forecasting tasks. Zhou et al. (2022) proposes two attention modules that operate in the frequency domain using Fourier or wavelet transformation.

**Orthogonal Basis and Neural Network**   Orthogonal basis project arbitrary functions onto a certain space and thus enable the representation learning in another view. Orthogonal basis family is easy to be discretized and serves as a plug-in operation in neural networks. Recent studies began to realize the efficiency and effectiveness of the Orthogonal basis, including the polynomial family and others (Fourier basis & Multiwavelet basis). Fourier basis is first introduced for acceleration due to the Fast Fourier algorithm, for example, acceleration of computing convolution Gu et al. (2020) or Auto-correlation function Wen et al. (2021a); Wu et al. (2021). Fourier basis also serves as a performance boosting block: Fourier with Recurrent structure Zhang et al. (2018), Fourier with MLP Li et al. (2020); Lee-Thorp et al. (2021) and Fourier in Transformer Zhou et al. (2022). Multiwavelet transform is a more local filter (compared with Fourier) and a frequency decomposer. Thus, neural networks which employ multiwavelet filter usually exhibit a hierarchical structure and treat different frequency in different tunnels, e.g., Wang et al. (2018); Gupta et al. (2021); Zhou et al. (2022). Orthogonal Polynomials are naturally good selections of orthogonal basis. Legendre Memory Unit (LMU) Voelker et al. (2019) uses Legendre Polynomials for an orthogonal projection of input signals for memory strengthening with the backbone of LSTM. The projection process is mathematically derived from delayed linear transfer function. HiPPO Gu et al. (2020), based on LMU, proposes a novel mechanism (Scaled Legendre), which involves the function's full history (LMU uses rolling windowed history). In the subsequent work of HiPPO, the authors propose S4 model Gu et al. (2021a) and give the first practice on time series forecasting tasks. However, LMU and HiPPO share the same backbone (LSTM), which may limit their performance.

# B Algorithm Implementation

**Algorithm 1** Frequency Enhanced Layer

```python
class Freq_enhanced_layer(nn.Module):
    def __init__(self, in_channels, out_channels, modes1, modes2, compression=0):
        super(Freq_enhanced_layer, self).__init__()
        self.in_channels = in_channels
        self.out_channels = out_channels
        self.modes1 = modes1 #Number of Fourier modes to multiply, at most floor(N/2) + 1
        self.modes2 = modes2
        self.compression = compression
        self.scale = (1 / (in_channels * out_channels))
        self.weights1 = nn.Parameter(self.scale * torch.rand(in_channels, out_channels, self.modes1))
        if compression>0: ## Low-rank approximation
            self.weights0 = nn.Parameter(self.scale * torch.rand(in_channels, self.compression, dtype=
                torch.cfloat))
            self.weights1 = nn.Parameter(self.scale * torch.rand(self.compression, self.compression, len
                (self.index), dtype=torch.cfloat))
            self.weights2 = nn.Parameter(self.scale * torch.rand(self.compression, out_channels, dtype=
                torch.cfloat))

    def forward(self, x):
        B, H,E, N = x.shape
        # Compute Fourier coefficients up to factor of e^(- something constant)
        x_ft = torch.fft.rfft(x)
        # Multiply relevant Fourier modes
        out_ft = torch.zeros(B, H, self.out_channels, x.size(-1)//2 + 1)
        if self.compression == 0:
            a = x_ft[:, :, :, :self.modes1]
            out_ft[:, :, :, :self.modes1] = torch.einsum("bjix,iox->bjox", a, self.weights1)
        else:
            a = x_ft[:, :, :, :self.modes2]
            a = torch.einsum("bjix,ih->bjhx", a, self.weights0)
            a = torch.einsum("bjhx,hkx->bjkx", a, self.weights1)
            out_ft[:, :, :, :self.modes2] = torch.einsum("bjkx,ko->bjox", a, self.weights2)
        # Return to physical space
        x = torch.fft.irfft(out_ft, n=x.size(-1))
        return x
```

**Algorithm 2** LPU layer

```python
from scipy import signal
from scipy import special as ss
class LPU(nn.Module):
    def __init__(self, N=256, dt=1.0, discretization='bilinear'):
        # N: the order of the Legendre projection
        # dt: step size - can be roughly inverse to the length of the sequence
        super(LPU,self).__init__()
        self.N = N
        A,B = transition(N) ### LMU projection matrix
        A,B, _, _, _ = signal.cont2discrete((A, B, C, D), dt=dt, method=discretization)
        B = B.squeeze(-1)
        self.register_buffer('A', torch.Tensor(A))
        self.register_buffer('B', torch.Tensor(B))
    def forward(self, inputs):
        # inputs: (length, ...)
        # output: (length, ..., N) where N is the order of the Legendre projection
        c = torch.zeros(inputs.shape[:-1] + tuple([self.N]))
        cs = []
        for f in inputs.permute([-1, 0, 1]):
            f = f.unsqueeze(-1)
            new = f @ self.B.unsqueeze(0) # [B, D, H, 256]
            c = F.linear(c, self.A) + new
            cs.append(c)
        return torch.stack(cs, dim=0)
    def reconstruct(self, c):
        a = (self.eval_matrix @ c.unsqueeze(-1)).squeeze(-1)
        return (self.eval_matrix @ c.unsqueeze(-1)).squeeze(-1)
```

# C   Dataset and Implementation Details

## C.1   Dataset Details

In this subsection, we summarize the details of the datasets used in this paper as follows: 1) ETT Zhou et al. (2021) dataset contains two sub-dataset: ETT1 and ETT2, collected from two separated counties. Each of them has two versions of sampling resolutions (15min & 1h). ETT dataset contains multiple time series of electrical loads and one time sequence of oil temperature. 2) Electricity dataset contains

---

https://archive.ics.uci.edu/ml/datasets/ElectricityLoadDiagrams 20112014

Table 8: Details of benchmark datasets.

| DATASET | LENGTH | DIMENSION | FREQUENCY |
|---------|--------|-----------|-----------|
| ETTM2 | 69680 | 8 | 15 MIN |
| EXCHANGE | 7588 | 9 | 1 DAY |
| WEATHER | 52696 | 22 | 10 MIN |
| ELECTRICITY | 26304 | 322 | 1H |
| ILI | 966 | 8 | 7 DAYS |
| TRAFFIC | 17544 | 863 | 1H |

the electricity consumption for more than three hundred clients with each column corresponding to one client. 3) Exchange Lai et al. (2018) dataset contains the current exchange of eight countries. 4) Traffic dataset contains the occupation rate of freeway systems in California, USA. 5) Weather dataset contains 21 meteorological indicators for a range of one year in Germany. 6) Illness dataset contains the influenza-like illness patients in the United States. Table 8 summarizes all the features of the six benchmark datasets. They are all split into the training set, validation set, and test set by the ratio of 7:1:2 during modeling.

## C.2 Implementation Details

Our model is trained using ADAM Kingma & Ba (2017) optimizer with a learning rate of $1e^{-4}$ to $1e^{-3}$. The batch size is set to 32 (It depends on the GPU memory used in the experiment. In fact, a batch size up to 256 does not deteriorate the performance but with faster training speed if larger memory GPU or multiple GPUs is used). The default training process is 15 epochs without any early stopping. We save the model with the lowest loss on the validation set for the final testing. The mean square error (MSE) and mean absolute error (MAE) are used as metrics. All experiments are repeated 5 times and the mean of the metrics is reported as the final results. All the deep learning networks are implemented using PyTorch Paszke et al. (2019) and trained on NVIDIA V100 32GB GPUs/NVIDIA V100 16GB GPUs.

## C.3 Experiment Error Bars

We train our model 5 times and calculate the error bars for FiLM and SOTA model FEDformer to compare the robustness, which is summarized in Table 9. It can be seen that the overall performance of the proposed FiLM is better than that of the SOTA FEDformer model.

Table 9: MSE with error bars (Mean and STD) for FiLM and FEDformer baseline for multivariate long-term forecasting. All experiments are repeated 5 times.

| MSE | | ETTm2 | Electricity | Exchange | Traffic |
|-----|-----|-------|-------------|----------|---------|
| FiLM | 96 | $0.165 \pm 0.0051$ | $0.153 \pm 0.0014$ | $0.079 \pm 0.002$ | $0.416 \pm 0.010$ |
| | 192 | $0.222 \pm 0.0038$ | $0.165 \pm 0.0023$ | $0.159 \pm 0.011$ | $0.408 \pm 0.007$ |
| | 336 | $0.277 \pm 0.0021$ | $0.186 \pm 0.0018$ | $0.270 \pm 0.018$ | $0.425 \pm 0.007$ |
| | 720 | $0.371 \pm 0.0066$ | $0.236 \pm 0.0022$ | $0.536 \pm 0.026$ | $0.520 \pm 0.003$ |
| FED-f | 96 | $0.203 \pm 0.0042$ | $0.194 \pm 0.0008$ | $0.148 \pm 0.002$ | $0.217 \pm 0.008$ |
| | 192 | $0.269 \pm 0.0023$ | $0.201 \pm 0.0015$ | $0.270 \pm 0.008$ | $0.604 \pm 0.004$ |
| | 336 | $0.325 \pm 0.0015$ | $0.215 \pm 0.0018$ | $0.460 \pm 0.016$ | $0.621 \pm 0.006$ |
| | 720 | $0.421 \pm 0.0038$ | $0.246 \pm 0.0020$ | $1.195 \pm 0.026$ | $0.626 \pm 0.003$ |

## C.4 Univariate Forecasting Results

The univariate benchmark results are summarized in Table 10.

## C.5 ETT Full Benchmark

We present the full-benchmark on four ETT datasets Zhou et al. (2021) in Table 11 (multivariate forecasting) and Table 12 (univariate forecasting). The ETTh1 and ETTh2 are recorded hourly while

---

http://pems.dot.ca.gov

https://www.bgc-jena.mpg.de/wetter/

https://gis.cdc.gov/grasp/fluview/fluportaldashboard.html

Table 10: Univariate long-term forecasting results on six datasets with various input length and prediction horizon $O \in \{96, 192, 336, 720\}$. A lower MSE indicates better performance. All experiments are repeated 5 times.

| Methods | | FiLM | | FEDformer | | Autoformer | | S4 | | Informer | | LogTrans | | Reformer | |
|---|---|---|---|---|---|---|---|---|---|---|---|---|---|---|---|
| Metric | | MSE | MAE | MSE | MAE | MSE | MAE | MSE | MAE | MSE | MAE | MSE | MAE | MSE | MAE |
| ETTm2 | 96 | 0.065 | **0.189** | **0.063** | 0.189 | 0.065 | 0.189 | 0.153 | 0.318 | 0.088 | 0.225 | 0.075 | 0.208 | 0.076 | 0.214 |
| | 192 | **0.094** | **0.233** | 0.102 | 0.245 | 0.118 | 0.256 | 0.183 | 0.350 | 0.132 | 0.283 | 0.129 | 0.275 | 0.132 | 0.290 |
| | 336 | **0.124** | **0.274** | 0.130 | 0.279 | 0.154 | 0.305 | 0.204 | 0.367 | 0.180 | 0.336 | 0.154 | 0.302 | 0.160 | 0.312 |
| | 720 | **0.173** | **0.323** | 0.178 | 0.325 | 0.182 | 0.335 | 0.482 | 0.567 | 0.300 | 0.435 | 0.160 | 0.321 | 0.168 | 0.335 |
| Electricity | 96 | **0.154** | **0.247** | 0.253 | 0.370 | 0.341 | 0.438 | 0.351 | 0.452 | 0.484 | 0.538 | 0.288 | 0.393 | 0.274 | 0.379 |
| | 192 | **0.166** | **0.258** | 0.282 | 0.386 | 0.345 | 0.428 | 0.373 | 0.455 | 0.557 | 0.558 | 0.432 | 0.483 | 0.304 | 0.402 |
| | 336 | **0.188** | **0.283** | 0.346 | 0.431 | 0.406 | 0.470 | 0.408 | 0.477 | 0.636 | 0.613 | 0.430 | 0.483 | 0.370 | 0.448 |
| | 720 | **0.249** | **0.341** | 0.422 | 0.484 | 0.565 | 0.581 | 0.472 | 0.517 | 0.819 | 0.682 | 0.491 | 0.531 | 0.460 | 0.511 |
| Exchange | 96 | **0.110** | **0.259** | 0.131 | 0.284 | 0.241 | 0.387 | 0.344 | 0.482 | 0.591 | 0.615 | 0.237 | 0.377 | 0.298 | 0.444 |
| | 192 | **0.207** | **0.352** | 0.277 | 0.420 | 0.300 | 0.369 | 0.362 | 0.494 | 1.183 | 0.912 | 0.738 | 0.619 | 0.777 | 0.719 |
| | 336 | **0.327** | **0.461** | 0.426 | 0.511 | 0.509 | 0.524 | 0.499 | 0.594 | 1.367 | 0.984 | 2.018 | 1.070 | 1.832 | 1.128 |
| | 720 | **0.811** | **0.708** | 1.162 | 0.832 | 1.260 | 0.867 | 0.552 | 0.614 | 1.872 | 1.072 | 2.405 | 1.175 | 1.203 | 0.956 |
| Traffic | 96 | **0.144** | **0.215** | 0.170 | 0.263 | 0.246 | 0.346 | 0.194 | 0.290 | 0.257 | 0.353 | 0.226 | 0.317 | 0.313 | 0.383 |
| | 192 | **0.120** | **0.199** | 0.173 | 0.265 | 0.266 | 0.370 | 0.172 | 0.272 | 0.299 | 0.376 | 0.314 | 0.408 | 0.386 | 0.453 |
| | 336 | **0.128** | **0.212** | 0.178 | 0.266 | 0.263 | 0.371 | 0.178 | 0.278 | 0.312 | 0.387 | 0.387 | 0.453 | 0.423 | 0.468 |
| | 720 | **0.153** | **0.252** | 0.187 | 0.286 | 0.269 | 0.372 | 0.263 | 0.386 | 0.366 | 0.436 | 0.491 | 0.437 | 0.378 | 0.433 |
| Weather | 96 | **0.0012** | **0.026** | 0.0035 | 0.046 | 0.011 | 0.081 | 0.0061 | 0.065 | 0.0038 | 0.044 | 0.0046 | 0.052 | 0.012 | 0.087 |
| | 192 | **0.0014** | **0.029** | 0.0054 | 0.059 | 0.0075 | 0.067 | 0.0067 | 0.067 | 0.0023 | 0.040 | 0.0056 | 0.060 | 0.0098 | 0.079 |
| | 336 | **0.0015** | **0.030** | 0.0041 | 0.050 | 0.0063 | 0.062 | 0.0025 | 0.0381 | 0.0041 | 0.049 | 0.0060 | 0.054 | 0.0050 | 0.059 |
| | 720 | **0.0022** | **0.037** | 0.015 | 0.091 | 0.0085 | 0.070 | 0.0074 | 0.0736 | 0.0031 | 0.042 | 0.0071 | 0.063 | 0.0041 | 0.049 |
| ILI | 24 | **0.629** | **0.538** | 0.693 | 0.629 | 0.948 | 0.732 | 0.866 | 0.584 | 5.282 | 2.050 | 3.607 | 1.662 | 3.838 | 1.720 |
| | 36 | **0.444** | **0.481** | 0.554 | 0.604 | 0.634 | 0.650 | 0.622 | 0.532 | 4.554 | 1.916 | 2.407 | 1.363 | 2.934 | 1.520 |
| | 48 | **0.557** | **0.584** | 0.699 | 0.696 | 0.791 | 0.752 | 0.813 | 0.679 | 4.273 | 1.846 | 3.106 | 1.575 | 3.754 | 1.749 |
| | 60 | **0.641** | **0.644** | 0.828 | 0.770 | 0.874 | 0.797 | 0.931 | 0.747 | 5.214 | 2.057 | 3.698 | 1.733 | 4.162 | 1.847 |

ETTm1 and ETTm2 are recorded every 15 minutes. The time series in ETTh1 and ETTm1 follow the same pattern, and the only difference is the sampling rate, similarly for ETTh2 and ETTm2. On average, our FiLM yields a **14.0%** relative MSE reduction for multivariate forecasting, and a **16.8%** reduction for univariate forecasting over the SOTA results from FEDformer.

Table 11: Multivariate long-term forecasting results on ETT full benchmark. The best results are highlighted in bold. A lower MSE indicates better performance. All experiments are repeated 5 times.

| Methods | | FiLM | | FEDformer | | Autoformer | | S4 | | Informer | | LogTrans | | Reformer | |
|---|---|---|---|---|---|---|---|---|---|---|---|---|---|---|---|
| Metric | | MSE | MAE | MSE | MAE | MSE | MAE | MSE | MAE | MSE | MAE | MSE | MAE | MSE | MAE |
| ETTh1 | 96 | **0.371** | **0.394** | 0.376 | 0.419 | 0.449 | 0.459 | 0.949 | 0.777 | 0.865 | 0.713 | 0.878 | 0.740 | 0.837 | 0.728 |
| | 192 | **0.414** | **0.423** | 0.420 | 0.448 | 0.500 | 0.482 | 0.882 | 0.745 | 1.008 | 0.792 | 1.037 | 0.824 | 0.923 | 0.766 |
| | 336 | **0.442** | **0.445** | 0.459 | 0.465 | 0.521 | 0.496 | 0.965 | 0.75 | 1.107 | 0.809 | 1.238 | 0.932 | 1.097 | 0.835 |
| | 720 | **0.465** | **0.472** | 0.506 | 0.507 | 0.514 | 0.512 | 1.074 | 0.814 | 1.181 | 0.865 | 1.135 | 0.852 | 1.257 | 0.889 |
| ETTh2 | 96 | **0.284** | **0.348** | 0.346 | 0.388 | 0.358 | 0.397 | 1.551 | 0.968 | 3.755 | 1.525 | 2.116 | 1.197 | 2.626 | 1.317 |
| | 192 | **0.357** | **0.400** | 0.429 | 0.439 | 0.456 | 0.452 | 2.336 | 1.229 | 5.602 | 1.931 | 4.315 | 1.635 | 11.12 | 2.979 |
| | 336 | **0.377** | **0.417** | 0.482 | 0.480 | 0.482 | 0.486 | 2.801 | 1.259 | 4.721 | 1.835 | 1.124 | 1.604 | 9.323 | 2.769 |
| | 720 | **0.439** | **0.456** | 0.463 | 0.474 | 0.515 | 0.511 | 2.973 | 1.333 | 3.647 | 1.625 | 3.188 | 1.540 | 3.874 | 1.697 |
| ETTm1 | 96 | **0.302** | **0.345** | 0.378 | 0.418 | 0.505 | 0.475 | 0.640 | 0.584 | 0.672 | 0.571 | 0.600 | 0.546 | 0.538 | 0.528 |
| | 192 | **0.338** | **0.368** | 0.426 | 0.441 | 0.553 | 0.496 | 0.570 | 0.555 | 0.795 | 0.669 | 0.837 | 0.700 | 0.658 | 0.592 |
| | 336 | **0.373** | **0.388** | 0.445 | 0.459 | 0.621 | 0.537 | 0.795 | 0.691 | 1.212 | 0.871 | 1.124 | 0.832 | 0.898 | 0.721 |
| | 720 | **0.420** | **0.420** | 0.543 | 0.490 | 0.671 | 0.561 | 0.738 | 0.655 | 1.166 | 0.823 | 1.153 | 0.820 | 1.102 | 0.841 |
| ETTm2 | 96 | **0.165** | **0.256** | 0.203 | 0.287 | 0.255 | 0.339 | 0.705 | 0.690 | 0.365 | 0.453 | 0.768 | 0.642 | 0.658 | 0.619 |
| | 192 | **0.222** | **0.296** | 0.269 | 0.328 | 0.281 | 0.340 | 0.924 | 0.692 | 0.533 | 0.563 | 0.989 | 0.757 | 1.078 | 0.827 |
| | 336 | **0.277** | **0.333** | 0.325 | 0.366 | 0.339 | 0.372 | 1.364 | 0.877 | 1.363 | 0.887 | 1.334 | 0.872 | 1.549 | 0.972 |
| | 720 | **0.371** | **0.389** | 0.421 | 0.415 | 0.422 | 0.419 | 2.074 | 1.074 | 3.379 | 1.338 | 3.048 | 1.328 | 2.631 | 1.242 |

# D   Low-rank Approximation for FEL

With the low-rank approximation of learnable matrix in Fourier Enhanced Layer significantly reducing our parameter size, here we study its effect on model accuracy on two typical datasets as shown in Table 13.

Table 12: Univariate long-term forecasting results on ETT full benchmark. The best results are highlighted in bold. A lower MSE indicates better performance. All experiments are repeated 5 times.

| Methods | FiLM | | FEDformer | | Autoformer | | S4 | | Informer | | LogTrans | | Reformer | |
|---|---|---|---|---|---|---|---|---|---|---|---|---|---|---|
| Metric | MSE | MAE | MSE | MAE | MSE | MAE | MSE | MAE | MSE | MAE | MSE | MAE | MSE | MAE |
| ETTh1 96 | **0.055** | **0.178** | 0.079 | 0.215 | 0.071 | 0.206 | 0.316 | 0.490 | 0.193 | 0.377 | 0.283 | 0.468 | 0.532 | 0.569 |
| ETTh1 192 | **0.072** | **0.207** | 0.104 | 0.245 | 0.114 | 0.262 | 0.345 | 0.516 | 0.217 | 0.395 | 0.234 | 0.409 | 0.568 | 0.575 |
| ETTh1 336 | **0.083** | **0.229** | 0.119 | 0.270 | 0.107 | 0.258 | 0.825 | 0.846 | 0.202 | 0.381 | 0.386 | 0.546 | 0.635 | 0.589 |
| ETTh1 720 | **0.090** | **0.240** | 0.127 | 0.280 | 0.126 | 0.283 | 0.190 | 0.355 | 0.183 | 0.355 | 0.475 | 0.628 | 0.762 | 0.666 |
| ETTh2 96 | **0.127** | 0.272 | 0.128 | **0.271** | 0.153 | 0.306 | 0.381 | 0.501 | 0.213 | 0.373 | 0.217 | 0.379 | 1.411 | 0.838 |
| ETTh2 192 | **0.182** | 0.335 | 0.185 | **0.330** | 0.204 | 0.351 | 0.332 | 0.458 | 0.227 | 0.387 | 0.281 | 0.429 | 5.658 | 1.671 |
| ETTh2 336 | **0.204** | **0.367** | 0.231 | 0.378 | 0.246 | 0.389 | 0.655 | 0.670 | 0.242 | 0.401 | 0.293 | 0.437 | 4.777 | 1.582 |
| ETTh2 720 | **0.241** | **0.396** | 0.278 | 0.420 | 0.268 | 0.409 | 0.630 | 0.662 | 0.291 | 0.439 | 0.218 | 0.387 | 2.042 | 1.039 |
| ETTm1 96 | **0.029** | **0.127** | 0.033 | 0.140 | 0.056 | 0.183 | 0.651 | 0.733 | 0.109 | 0.277 | 0.049 | 0.171 | 0.296 | 0.355 |
| ETTm1 192 | **0.041** | **0.153** | 0.058 | 0.186 | 0.081 | 0.216 | 0.190 | 0.372 | 0.151 | 0.310 | 0.157 | 0.317 | 0.429 | 0.474 |
| ETTm1 336 | **0.053** | **0.175** | 0.071 | 0.209 | 0.076 | 0.218 | 0.428 | 0.581 | 0.427 | 0.591 | 0.289 | 0.459 | 0.585 | 0.583 |
| ETTm1 720 | **0.071** | **0.205** | 0.102 | 0.250 | 0.110 | 0.267 | 0.254 | 0.433 | 0.438 | 0.586 | 0.430 | 0.579 | 0.782 | 0.730 |
| ETTm2 96 | 0.065 | **0.189** | **0.063** | 0.189 | 0.065 | 0.189 | 0.153 | 0.318 | 0.088 | 0.225 | 0.075 | 0.208 | 0.076 | 0.214 |
| ETTm2 192 | **0.094** | **0.233** | 0.102 | 0.245 | 0.118 | 0.256 | 0.183 | 0.350 | 0.132 | 0.283 | 0.129 | 0.275 | 0.132 | 0.290 |
| ETTm2 336 | **0.124** | **0.274** | 0.130 | 0.279 | 0.154 | 0.305 | 0.204 | 0.367 | 0.180 | 0.336 | 0.154 | 0.302 | 0.160 | 0.312 |
| ETTm2 720 | **0.173** | **0.323** | 0.178 | 0.325 | 0.182 | 0.335 | 0.482 | 0.567 | 0.300 | 0.435 | 0.160 | 0.321 | 0.168 | 0.335 |

Table 13: Low-rank Approximation (LRA) study for frequency enhanced layer: Comp. K=0 means default version without LRA, 1 means the largest compression using K=1.

| Comp. K | 0 | | 16 | | 4 | | 1 | |
|---|---|---|---|---|---|---|---|---|
| Metric | MSE | MAE | MSE | MAE | MSE | MAE | MSE | MAE |
| ETTh1 96 | **0.371** | **0.394** | 0.371 | 0.397 | 0.373 | 0.399 | 0.391 | 0.418 |
| ETTh1 192 | 0.414 | **0.423** | 0.414 | 0.425 | **0.413** | 0.426 | 0.437 | 0.445 |
| ETTh1 336 | **0.442** | 0.445 | 0.452 | 0.451 | 0.445 | **0.444** | 0.460 | 0.458 |
| ETTh1 720 | **0.454** | **0.451** | 0.460 | 0.472 | 0.461 | 0.471 | 0.464 | 0.476 |
| Weather 96 | **0.199** | **0.262** | 0.200 | 0.266 | 0.199 | 0.263 | 0.198 | 0.261 |
| Weather 192 | 0.228 | 0.288 | 0.232 | 0.298 | 0.227 | 0.287 | **0.226** | **0.285** |
| Weather 336 | 0.267 | 0.323 | 0.266 | 0.320 | **0.253** | **0.314** | 0.264 | 0.316 |
| Weather 720 | 0.319 | 0.361 | **0.314** | **0.352** | 0.319 | 0.361 | 0.314 | 0.354 |
| Parameter size | 100% | | 6.4% | | 1.6% | | 0.4% | |

# E    Theoretical Analysis

## E.1    Theorem 1

The proof is a simple extension of Proposition 6 in Gu et al. (2020). We omit it for brevity.

## E.2    Theorem 2

As we have $x_t = Ax_{t-1} + b - \epsilon_{t-1}$ for $t = 2, 3, ...\theta$, we recursively use them and the following result holds:

$$\begin{aligned}
x_t &= Ax_{t-1} + b + \epsilon_{t-1} \\
&= A(Ax_{t-2} + b + \epsilon_{t-2}) + b + \epsilon_{t-1} \\
&= A^2 x_{t-2} + Ab + b + A\epsilon_{t-2} + \epsilon_{t-1} \\
&\cdots \\
&= A^\theta x_{t-\theta} + \sum_{i=1}^{\theta-1} A^i b + \underbrace{\sum_{i=1}^{\theta-1} A^i \epsilon_{t-i}}_{(*)}.
\end{aligned}$$

Following Hoeffding inequality, for $\mu > 0$ we have

$$\mathbb{P}\left(|(*)| \geq \mu\right) \leq \exp\left(-\frac{2\mu^2}{\sum_{i=1}^{\theta-1} \|A^i \epsilon_{t-1}\|_{\psi_2}^2}\right), \tag{4}$$

where $\| \cdot \|_{\psi_2}$ is the Orlicz norm defined as

$$\|X\|_{\psi_2} := \inf \left\{ c \geq 0 : \mathbb{E}[\exp(X^2/c^2)] \leq 2 \right\}.$$

Since we require $A$ being unitary, we will have $\|A\epsilon\|_2^2 = \|\epsilon\|_2^2$ and it implies $\|A^i\epsilon_{t-1}\|_{\psi_2}^2\| = \|\epsilon_{t-1}\|_{\psi_2}^2 = \mathcal{O}(\sigma^2)$ for $i = 1, 2, ..., \theta$. The desirable result follows by setting $\mu = \mathcal{O}(\sqrt{\theta}\sigma)$.

### E.3 Theorem 3

As we keep first $s$ columns selected, $P(A) - A$ has all 0 elements in first $s$ columns. We thus ignore them and consider the approximation quality on $\tilde{A} \in \mathbb{R}^{d \times (n-s)}$ with the sampled columns. Via the similar analysis in Appendix C of Zhou et al. (2022), with high probability we have $\|\tilde{A} - P(A)\|_F \leq (1 + \epsilon)\|\tilde{A} - \tilde{A}_k\|_F$, where $\tilde{A}_k$ is the "best" rank-$k$ approximation provided by truncating the singular value decomposition (SVD) of $\tilde{A}$, and where $\| \cdot \|_F$ is the Frobenius norm. As we assume the element in last $n - s$ columns of $A$ is smaller than $a_{\min}$, one can verify $\|\tilde{A} - P(A)\|_F \leq \sqrt{d \times (n-s)} a_{\min}$ and desirable result follows immediately.

## F Parameter Sensitivity

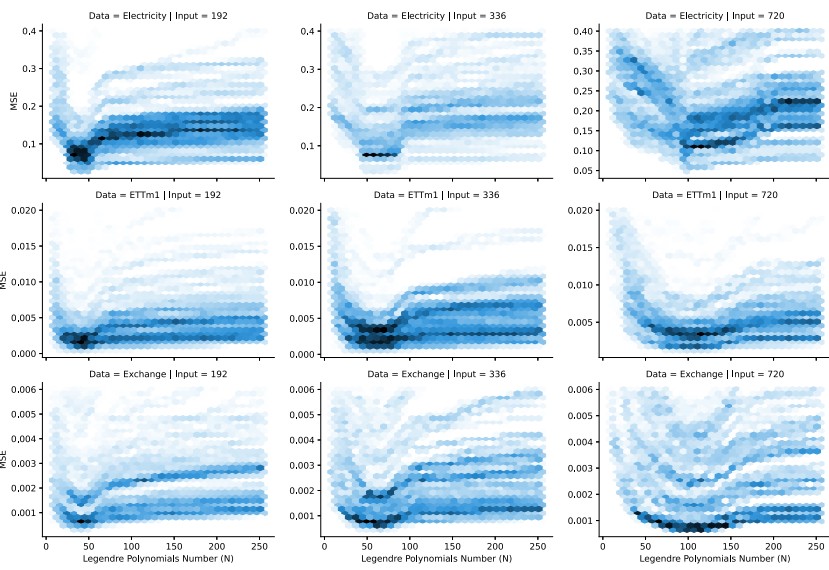

Figure 7: The reconstruction error (MSE) vs. Legendre Polynomial number ($N$) on three datasets with three different input lengths.

**Influence of Legendre Polynomial number $N$ and Frequency mode number $M$** The experimental results on three different datasets (ETTm1, Electricity, and Exchange) in Figure 7 show the optimal choice of Legendre Polynomials number ($N$) when we aim to minimize the reconstruction error (in MSE) on the historical data. The MSE error decreases sharply at first and saturates at an optimal point, where $N$ is in proportion to the input length. For input sequences with lengths of 192, 336, and 720, $N \approx 40$, 60, and 100 gives the minimal MSE, respectively.

Figure 8 shows the MSE error of time series forecasting on the Electricity dataset, with different Legendre Polynomials number ($N$), mode number, and input length. We observe that, when enlarging $N$, the model performance saturates at an optimal point. For example, in Figure 8 Left (input length=192), the best performance is reached when $N > 64$. While in Figure 8 Right (input length=720), the best performance is reached when $N$ is larger than 128. Another influential parameter is the mode number. From Figure 8 we observe that a small mode number will lead to better performance, as a module with a small mode number works as a denoising filter.

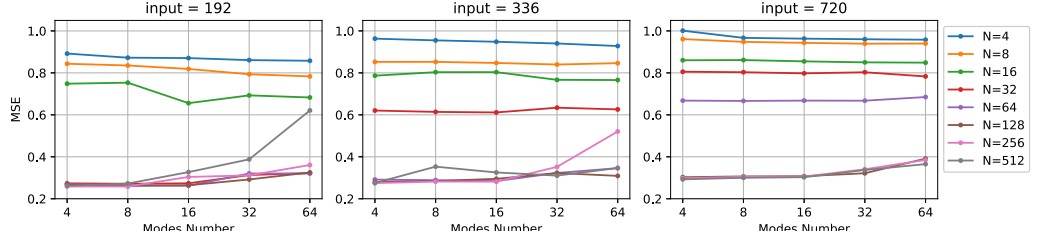

Figure 8: The MSE error of univariate time series forecasting task on Electricity dataset with different Legendre Polynomials number ($N$), mode number and input length. Left: input length = 192. Mid: input length = 336. Right: input length = 720.

# G  Noise Injection Experiment

Our model's robustness in long-term forecasting tasks can be demonstrated using a series of noise injection experiments as shown in Table 14. As we can see, adding Gaussian noise in the training/test stage has a limited effect on our model's performance, since the deterioration is less than 1.5% in the worst case. The model's robustness is consistent across various forecasting horizons. Note that adding the noise in the testing stage other than the training stage will even improve our performance by 0.4%, which further supports our claim of robustness.

Table 14: Noise injection studies. A 0.3*$\mathcal{N}(0,1)$ Gaussian noise is introduced into our training/testing. We conduct 4 sets of experiments with/without noise in training and test phases. The experiments are performed on ETTm1 and Electricity with different output lengths. The metric of variants is presented in relative value ('+' indicates degraded performance, and '-' indicates improved performance).

| Training | | noise | | | with noise | | | |
|---|---|---|---|---|---|---|---|---|
| Testing | without noise | | with noise | | without noise | | with noise | |
| Metric | MSE | MAE | MSE | MAE | MSE | MAE | MSE | MAE |
| 96 | **0.371** | **0.394** | -1.6% | -2.0% | -0.0% | -0.0% | -1.6% | -2.0% |
| 192 | **0.414** | **0.423** | -0.5% | -1.4% | +0.5% | -0.0% | -0.5% | -1.4% |
| 336 | **0.442** | **0.445** | -1.8% | -0.9% | -0.9% | +1.3% | -3.2% | -1.6% |
| 720 | **0.465** | **0.472** | +0.2% | -0.2% | +0.9% | +0.9% | -0.6% | -0.4% |

(row label: *ETTh1*)

# H  Distribution Analysis of Forecasting Output

## H.1  Kolmogorov-Smirnov Test

We adopt the Kolmogorov-Smirnov (KS) test to check whether our model's input and output sequences come from the same distribution. The KS test is a nonparametric test for checking the equality of probability distributions. In essence, the test answers the following question "Are these two sets of samples drawn from the same distribution?". The Kolmogorov-Smirnov statistic is defined as:

$$D_{n,m} = \sup_x |F_{1,n}(x) - F_{2,m}(x)|,$$

where $\sup$ is the supremum function, $F_{1,n}$ and $F_{2,m}$ are the empirical distribution functions of the two compared samples. For samples that are large enough, the null hypothesis would be rejected at level $\alpha$ if

$$D_{n,m} > \sqrt{-\frac{1}{2}\ln\left(\frac{\alpha}{2}\right)} \cdot \sqrt{\frac{n+m}{n \cdot m}},$$

where $n$ and $m$ are the first and second sample sizes.

## H.2  Distribution Analysis

In this section, we evaluate the distribution similarity between models' input and output sequences using the KS test. In Table 15, we applied the Kolmogrov-Smirnov test to check if the output

sequences of various models that trained on ETTm1/ETTm2 are consistent with the input sequence. On both datasets, by setting the standard P-value as 0.01, various existing baseline models have much smaller P-values except FEDformer and Autoformer, which indicates their outputs have a high probability of being sampled from different distributions compared to their input signals. Autoformer and FEDformer have much larger P-values mainly due to their seasonal-trend decomposition mechanism. The proposed FiLM also has a much larger P-value compared to most baseline models. And its null hypothesis can not be rejected in most cases for these two datasets. It implies that the output sequence generated by FiLM shares a similar pattern as the input signal, and thus justifies our design motivation of FiLM as discussed in Section 1. Though FiLM gets a smaller P-value than FEDformer, it is close to the actual output, which indicates that FiLM makes a good balance between recovering and forecasting.

Table 15: P-values of Kolmogrov-Smirnov test of different Transformer models for long-term forecasting output on ETTm1 and ETTm2 dataset. Larger value indicates the hypothesis (the input sequence and forecasting output come from the same distribution) is less likely to be rejected. The largest results are highlighted.

| Methods | | Transformer | Informer | Autoformer | FEDformer | FiLM | True |
|---|---|---|---|---|---|---|---|
| ETTm1 | 96 | 0.0090 | 0.0055 | 0.020 | **0.048** | 0.016 | 0.023 |
| | 192 | 0.0052 | 0.0029 | 0.015 | **0.028** | 0.0123 | 0.013 |
| | 336 | 0.0022 | 0.0019 | 0.012 | **0.015** | 0.0046 | 0.010 |
| | 720 | 0.0023 | 0.0016 | 0.008 | **0.014** | 0.0024 | 0.004 |
| ETTm2 | 96 | 0.0012 | 0.0008 | **0.079** | 0.071 | 0.022 | 0.087 |
| | 192 | 0.0011 | 0.0006 | **0.047** | 0.045 | 0.020 | 0.060 |
| | 336 | 0.0005 | 0.00009 | 0.027 | **0.028** | 0.012 | 0.042 |
| | 720 | 0.0008 | 0.0002 | **0.023** | 0.021 | 0.0081 | 0.023 |

# I    Learnable Parameter Size

Compared to Transformer-based baseline models, FiLM enjoys a lightweight property with **80%** learnable parameter reduction as shown in Table 16. It has the potential to be used in mobile devices, or, in some situations where a lightweight model is preferred.

Table 16: Parameter size of baseline models and FiLM with different low-rank approximations: the models are trained and tested on ETT dataset; the subscript number denotes $k$ in low-rank approximation.

| Methods | Transformer | Autoformer | FEDformer | FiLM | FiLM$_{16}$ | FiLM$_4$ | FiLM$_1$ |
|---|---|---|---|---|---|---|---|
| Parameter(M) | 0.0069 | 0.0069 | 0.0098 | 1.50 | 0.0293 | 0.0062 | 0.00149 |

# J    Training Speed and Memory Usage

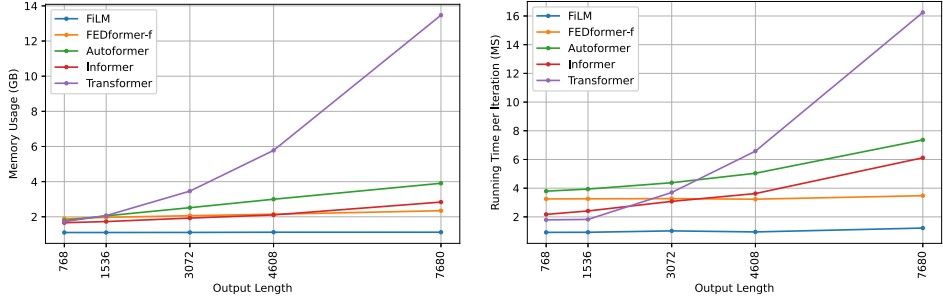

Figure 9: (Left) the memory usage of FiLM and baseline models. (Right) training speed of FiLM and baseline models. The input length is fixed to 96 and the output length is 768, 1536, 3072, 4608, and 7680.

**Memory Usage**    As shown in Figure 9 (Left), FiLM has good memory usage with the prolonging output length. For a fair comparison, we fix the experimental settings of *Xformer*, where we fix the input length as 96 and prolong the output length. From Figure 9 (Left), we can observe that FiLM

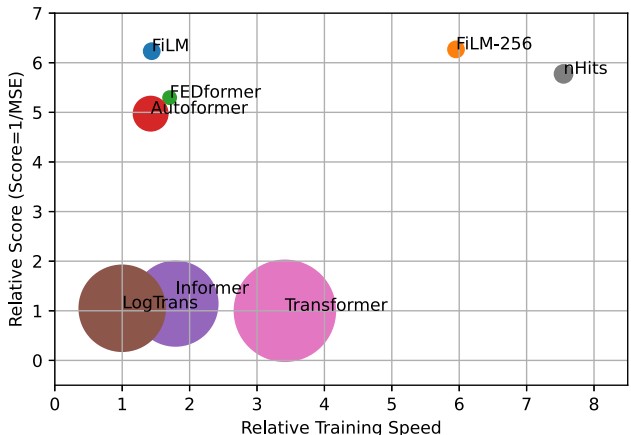

Figure 10: Comparison of training speed and performance of benchmarks. The experiment is performed on ETTm2 with output length = 96, 192, 336, and 720. The performance of the models is measured with *Score*, where $Score = 1/MSE$. The radius of the circle measured the STD of the performance. A higher *Score* indicates better performance, same for *Speed*. A smaller circle indicates better robustness. The *Speed* and *Score* are presented on relative value.

has a quasi-constant memory usage. Note that the memory usage of FiLM is only linear to the input length. Furthermore, FiLM enjoys a much smaller memory usage than others because of the simple architecture and compressed parameters with low-rank approximation as discussed in Appendix I.

**Training Speed** Experiments are performed on one NVIDIA V100 32GB GPU. As shown in Figure 9 (Right), FiLM has a faster training speed than others with the prolonging output length. For a fair comparison, we fix the experimental setting of *Xformer*, where we fix the input length as 96 and prolong the output length. However, in the real experiment settings, we use a longer input length (much longer than 96). Thus, the experiment in Figure 9 (Right) is merely a toy case to show the tendency. In Figure 10, we show the average epoch time vs average performance under the settings of benchmarks. The experiment is performed on ETTm2 dataset with output lengths = 96, 192, 336, and 720. Because of the extremely low memory usage of FiLM, it can be trained with a larger batch size (batch size = 256) on only one GPU compared with baselines (batch size = 32). In Figure 10, FiLM-256 is the FiLM models trained with batch size = 256, it exhibits significant advantages on both speed and accuracy. Furthermore, due to the shallow structure and smaller amount of trainable parameters, FiLM is easy to converge and enjoys smaller performance variation and smaller performance degradation when using a large batch size. It is observed that the models with Fourier enhanced block (FiLM & FEDformer) have better robustness. It is also worth noting that the vanilla Transformer has good training speed because of the not-so-long sequence length. Only a sequence length over one thousand will distinguish the advantage of efficient Transformers.

## K    Additional Benchmarks

### K.1    Multivariate long-term series forecasting with extra baseline models

For the additional benchmarks for multivariate experiments, we add some non-Transformer methods for comparison.N-BEATSOreshkin et al. (2019) and N-HiTSChallu et al. (2022) are two recent proposed powerful non-Transformer methods. As N-HiTS is the latest development from the research group, which also published N-BEATS, we add N-HiTS to our empirical comparison. Here, we adopt the results in the N-HiTS paper to prevent inappropriate parameter tuning problems. We also add a seasonal-naive model in the comparison. FiLM outperforms N-HiTS in most cases(33/48). Moreover, Simple Seasonal-naiveMakridakis et al. (1982) is a solid baseline on exchange datasets better than N-hits, Fedformer, and Autoformer, but FiLM still surpasses its performance, as shown in Table 17.

Table 17: multivariate long-term series forecasting results on six datasets with various input length and prediction length $O \in \{96, 192, 336, 720\}$ (For ILI dataset, we set prediction length $O \in \{24, 36, 48, 60\}$). Supplementary results of non-Transformer baselines (N-Hits and a seasonal-naive model).

| Methods | | FiLM | | N-Hits | | FEDformer | | Autoformer | | Seasonal-naive | |
|---|---|---|---|---|---|---|---|---|---|---|---|
| Metric | | MSE | MAE | MSE | MAE | MSE | MAE | MSE | MAE | MSE | MAE |
| ETTm2 | 96 | **0.165** | 0.256 | 0.176 | **0.255** | 0.203 | 0.287 | 0.255 | 0.339 | 0.262 | 0.300 |
| | 192 | **0.222** | **0.296** | 0.245 | 0.305 | 0.269 | 0.328 | 0.281 | 0.340 | 0.319 | 0.337 |
| | 336 | **0.277** | **0.333** | 0.295 | 0.346 | 0.325 | 0.366 | 0.339 | 0.372 | 0.375 | 0.371 |
| | 720 | **0.371** | **0.389** | 0.401 | 0.426 | 0.421 | 0.415 | 0.422 | 0.419 | 0.469 | 0.422 |
| Electricity | 96 | 0.154 | 0.267 | **0.147** | **0.249** | 0.183 | 0.297 | 0.201 | 0.317 | 0.211 | 0.278 |
| | 192 | **0.164** | **0.258** | 0.167 | 0.269 | 0.195 | 0.308 | 0.222 | 0.334 | 0.214 | 0.282 |
| | 336 | 0.188 | **0.283** | **0.186** | 0.290 | 0.212 | 0.313 | 0.231 | 0.338 | 0.226 | 0.294 |
| | 720 | 0.236 | **0.332** | 0.243 | 0.340 | **0.231** | 0.343 | 0.254 | 0.361 | 0.265 | 0.324 |
| Exchange | 96 | **0.086** | **0.204** | 0.092 | 0.211 | 0.139 | 0.276 | 0.197 | 0.323 | 0.086 | 0.204 |
| | 192 | **0.189** | **0.292** | 0.208 | 0.322 | 0.256 | 0.369 | 0.300 | 0.369 | 0.172 | 0.295 |
| | 336 | **0.356** | **0.433** | 0.371 | 0.443 | 0.426 | 0.464 | 0.509 | 0.524 | 0.311 | 0.401 |
| | 720 | **0.727** | **0.669** | 0.888 | 0.723 | 1.090 | 0.800 | 1.447 | 0.941 | 0.832 | 0.686 |
| Traffic | 96 | 0.416 | 0.294 | **0.402** | **0.282** | 0.562 | 0.349 | 0.613 | 0.388 | 1.219 | 0.497 |
| | 192 | **0.408** | **0.288** | 0.420 | 0.297 | 0.562 | 0.346 | 0.616 | 0.382 | 1.089 | 0.456 |
| | 336 | **0.425** | **0.298** | 0.448 | 0.313 | 0.570 | 0.323 | 0.622 | 0.337 | 1.147 | 0.473 |
| | 720 | **0.520** | **0.353** | 0.539 | 0.353 | 0.596 | 0.368 | 0.660 | 0.408 | 1.181 | 0.486 |
| Weather | 96 | 0.199 | 0.262 | **0.158** | **0.195** | 0.217 | 0.296 | 0.266 | 0.336 | 0.315 | 0.288 |
| | 192 | 0.228 | 0.288 | **0.211** | **0.247** | 0.276 | 0.336 | 0.307 | 0.367 | 0.341 | 0.305 |
| | 336 | **0.267** | 0.323 | 0.274 | **0.300** | 0.339 | 0.380 | 0.359 | 0.395 | 0.381 | 0.331 |
| | 720 | **0.319** | 0.361 | 0.351 | **0.353** | 0.403 | 0.428 | 0.578 | 0.578 | 0.440 | 0.370 |
| ILI | 24 | 1.970 | 0.875 | **1.862** | **0.869** | 2.203 | 0.963 | 3.483 | 1.287 | 6.581 | 1.699 |
| | 36 | **1.982** | **0.859** | 2.071 | 0.969 | 2.272 | 0.976 | 3.103 | 1.148 | 7.121 | 1.882 |
| | 48 | **1.868** | **0.896** | 2.346 | 1.042 | 2.209 | 0.981 | 2.669 | 1.085 | 6.567 | 1.797 |
| | 60 | **2.057** | **0.929** | 2.560 | 1.073 | 2.545 | 1.061 | 2.770 | 1.125 | 5.885 | 1.675 |

## K.2  Ablation univariate forecasting experiments for FEL layers with all six datasets

As shown in Table 18, although LPU+MLP combining all boosting tricks has slightly better performance than FiLM for the ETTm1 dataset, FiLM remains the most consistent and effective model among all variants across all six datasets. FEL is a much better backbone structure than MLP, LSTM, CNN, and vanilla attention modules.

## K.3  Boosting experiments of LPU with common deep learning backbones for all six datasets

As shown in Table 19, LPU shows a consistent boosting effect across all selected common deep learning backbones for most datasets. It can be used as a simple and effective build add-on block for long-term time series forecasting tasks. Although without data normalization, pure LPU negatively boosts performance for some cases.

## K.4  Ablation univariate forecasting experiments for Low rank approximation with all six datasets

As shown in Table 20, with the low-rank approximation of learnable matrix in the Fourier Enhanced Layer significantly reduces our parameter size, and even improves our model's performance for some datasets.

## K.5  Ablation univariate forecasting experiments for frequency mode selection with all six datasets

Three different mode selection policies are studied for frequency enhanced layer: 1) lowest mode selection: we select $m$ lowest frequency modes to retain. 2) random model selection: we select $m$ frequency modes randomly to retain. 3) lowest with extra high mode selection: we select $0.8 \times m$ lowest frequency modes and $0.2 \times m$ high-frequency modes randomly to retain. The experimental results are summarized in Table 21 with $m = 64$ for both experiments. The lowest mode selection is the most stable frequency mode selection policy through adding some randomness mode can improve the results for some datasets.

Table 18: : (Full Benchmark)Ablation studies of FEL layer. The FEL layer is replaced with 4 different variants: MLP, LSTM, CNN, and Transformer.The experiments are performed on ETTm1 and Electricity. The metric of variants is presented in relative value ('+' indicates degraded performance, '-' indicates improved performance).

| Methods | | FilM | | LPU+MLP | | LPU+LSTM | | LPU+CNN | | LPU+attention | |
|---|---|---|---|---|---|---|---|---|---|---|---|
| Metric | | MSE | MAE | MSE | MAE | MSE | MAE | MSE | MAE | MSE | MAE |
| ETTm1 | 96 | **0.029** | **0.127** | +0.0% | +0.0% | +12.1% | +7.1% | +13.5% | +9.5% | +1.7% | +1.6% |
| | 192 | **0.041** | **0.153** | -1.5% | -0.6% | +12.2% | +8.5% | +10.8% | +7.8% | +2.0% | +3.3% |
| | 336 | **0.053** | **0.175** | -1.7% | -1.7% | +4.5% | +4.0% | +10.8% | +6.3% | +4.5% | +2.9% |
| | 720 | **0.071** | **0.205** | -0.9% | -1.0% | +5.5% | +3.4% | +8.6% | +4.9% | +13.8% | +7.8% |
| Electricity | 96 | **0.154** | **0.247** | +155% | +81% | +160% | +84% | +330% | +155% | +242% | +119% |
| | 192 | **0.166** | **0.258** | +59% | +39% | +121% | +67% | +224% | +117% | +264% | +131% |
| | 336 | **0.188** | **0.283** | +55% | +35% | +150% | +74% | +128% | +71% | +183% | +95% |
| | 720 | **0.249** | **0.341** | +33% | +25% | +154% | +73% | +192% | +95% | +312% | +138% |
| Exchange | 96 | **0.110** | **0.247** | -13% | -12% | +51% | +17% | +4.6% | -1.2% | -4.6% | -5.8% |
| | 192 | **0.207** | **0.352** | +7.2% | +0.0% | +69% | +32% | +29% | +12% | +22% | +11% |
| | 336 | **0.327** | **0.461** | +48% | +13% | +62% | +20% | +68% | +24% | +72% | +23% |
| | 720 | **0.811** | **0.708** | +29% | +14% | +24% | +9.6% | +38% | +12% | +64% | +27% |
| Traffic | 96 | **0.144** | **0.215** | +69% | +47% | +13% | +15% | +300% | +176% | +271% | +161% |
| | 192 | **0.120** | **0.199** | +17% | +7.5% | +31% | +24% | +258% | +149% | +1572% | +355% |
| | 336 | **0.128** | **0.212** | +6.2% | +7.6% | +16% | +15% | +151% | +102% | +1514% | +368% |
| | 720 | **0.153** | **0.252** | +38% | +28% | +11% | +7.9% | +250% | +126% | +1048% | +349% |
| Weather | 96 | **0.0012** | **0.026** | +17% | +6.2% | +16% | +6.9% | +19% | +8.1% | +21% | +8.9% |
| | 192 | **0.0014** | **0.029** | -1.4% | -2.4% | +5.0% | +1.7% | 0.7% | -0.7% | +4.3% | +1.4% |
| | 336 | **0.0015** | **0.030** | +0.0% | -0.6% | +3.3% | +1.3% | +2.0% | +0.0% | +3.3% | +1.3% |
| | 720 | **0.0022** | **0.037** | +4.6% | -0.3% | +4.1% | -1.6% | 3.6% | 0.0% | +0.0% | -3.8% |
| ILI | 96 | **0.629** | **0.538** | +51% | +45% | -2.5% | 9.5% | +20% | +29% | +112% | +59% |
| | 192 | **0.444** | **0.481** | +99% | +58% | +25% | +24% | +84% | +56% | +360% | +142% |
| | 336 | **0.557** | **0.584** | +33% | +31% | +21% | +16% | +58% | +30% | +702% | +94% |
| | 720 | **0.641** | **0.644** | +8.4% | +5.4% | +23% | +18% | +42% | +22% | +74% | +34% |

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

Table 20: Low-rank Approximation (LRA) univariate forecasting study for frequency enhanced layer: Comp. K=0 means default version without LRA, 1 means the largest compression using K=1.

| Comp. K | | 0 | | 16 | | 4 | | 1 | |
|---|---|---|---|---|---|---|---|---|---|
| Metric | | MSE | MAE | MSE | MAE | MSE | MAE | MSE | MAE |
| ETTm2 | 96 | 0.065 | 0.189 | 0.064 | 0.185 | **0.064** | **0.185** | 0.064 | 0.186 |
| | 192 | 0.094 | 0.233 | 0.094 | 0.231 | 0.093 | 0.231 | **0.093** | **0.231** |
| | 336 | 0.124 | 0.274 | 0.124 | 0.270 | **0.124** | **0.269** | 0.124 | 0.271 |
| | 720 | 0.173 | 0.323 | 0.173 | 0.322 | **0.173** | **0.322** | 0.177 | 0.328 |
| Electricity | 96 | **0.154** | **0.247** | 0.211 | 0.324 | 0.216 | 0.331 | 0.277 | 0.387 |
| | 192 | **0.166** | **0.258** | 0.251 | 0.352 | 0.246 | 0.347 | 0.334 | 0.421 |
| | 336 | **0.188** | **0.283** | 0.276 | 0.369 | 0.302 | 0.396 | 0.363 | 0.440 |
| | 720 | **0.249** | **0.341** | 0.336 | 0.429 | 0.342 | 0.436 | 0.411 | 0.481 |
| Exchange | 96 | 0.110 | 0.259 | 0.119 | 0.273 | **0.104** | **0.247** | 0.105 | 0.251 |
| | 192 | 0.207 | 0.352 | 0.196 | 0.355 | **0.195** | **0.349** | 0.212 | 0.372 |
| | 336 | **0.327** | **0.461** | 0.388 | 0.497 | 0.373 | 0.491 | 0.407 | 0.506 |
| | 720 | **0.811** | **0.708** | 0.908 | 0.767 | 1.288 | 0.941 | 1.840 | 1.153 |
| Traffic | 96 | **0.144** | **0.215** | 0.146 | 0.223 | 0.154 | 0.237 | 0.267 | 0.373 |
| | 192 | **0.120** | **0.199** | 0.121 | 0.201 | 0.138 | 0.231 | 0.218 | 0.333 |
| | 336 | 0.128 | 0.212 | **0.120** | **0.206** | 0.132 | 0.227 | 0.216 | 0.335 |
| | 720 | **0.153** | **0.252** | 0.155 | 0.257 | 0.154 | 0.257 | 0.246 | 0.366 |
| Weather | 96 | 0.0012 | 0.026 | 0.0011 | 0.025 | **0.001** | **0.025** | 0.001 | 0.025 |
| | 192 | 0.0014 | 0.029 | 0.0014 | 0.028 | **0.001** | **0.028** | 0.001 | 0.028 |
| | 336 | 0.0015 | 0.03 | 0.0015 | 0.030 | **0.001** | **0.029** | 0.002 | 0.030 |
| | 720 | 0.0022 | 0.037 | 0.0022 | 0.037 | **0.002** | **0.037** | 0.002 | 0.037 |
| ILI | 96 | 0.629 | 0.538 | **0.599** | **0.556** | 0.628 | 0.558 | 0.630 | 0.579 |
| | 192 | **0.444** | **0.481** | 0.487 | 0.533 | 0.508 | 0.561 | 0.570 | 0.612 |
| | 336 | 0.557 | 0.584 | **0.553** | **0.565** | 0.703 | 0.696 | 0.722 | 0.706 |
| | 720 | **0.641** | **0.644** | 0.648 | 0.641 | 0.900 | 0.780 | 1.493 | 1.032 |
| Parameter size | | 100% | | 6.4% | | 1.6% | | 0.4% | |