# OpenReview forum: "FiLM: Frequency improved Legendre Memory Model for Long-term Time Series Forecasting"
_NeurIPS.cc/2022/Conference — NeurIPS 2022 Accept_

### Official Review · Reviewer_nNQa · 2022-07-10

**Rating:** 5
**Confidence:** 5
**Soundness:** 2 fair
**Presentation:** 3 good
**Contribution:** 3 good

**Summary:**

The paper proposes FiLM, a novel model based on Legendre projections, for long-horizon forecasting. The paper proposes two novel components: Legendre projection unit (LPU) and Fourier Enhanced Layer (FEL), which are combined in the FiLM architecture. Both LPU and FEL components can be used on multiple architectures. The authors tested the proposed approach on several benchmark datasets and compare it against recent Transformer-based models, and provide comprehensive ablation studies of the proposed components.

**Questions:**

- Why is the performance of FiLM different in ablation tables (table 2, 5) than from the main result table?
- Can you provide an explanation on how the model learns relations between time series or is the model univariate?
- How did you choose the hyperparameters of the model for the main table, in particular, M and N? The sensitivity analysis in appendix F shows some approximate optimal rules for N. However, these results are performed on the test set. Ideally, hyperparameters should be chosen based on the validation set (for example on a small grid over different values of M, N, etc).


**Limitations:**

Limitations are discussed, and I do not identify potential negative social impacts.

**Strengths And Weaknesses:**

Strengths:
- The paper proposes two original components: LPU and FEL, which can be incorporated by many different architectures.
- FiLM achieves superior performance than baselines, including the recent FED-former model, and the proposed components improve the performance of other architectures.
- Authors provide some theoretical results which support the design of the components.
- FiLM is a simpler and faster model than Transformer based models.
- The paper is well written and clear.
- Long-horizon forecasting is a very relevant topic and an active area of research.

Weaknesses:
- Recent studies have shown that many transformer-based models do not achieve SoTA performance in this setting [1, 2]. In many cases, improvements over previous models are caused by flawed comparisons (such as not tunning baselines properly, I confirmed with the authors of the Autoformer that, for example, they did not tune baselines such as the N-BEATS) or by omitting stronger baselines. Authors should include non-Transformer methods as well, including simple models. For example, a simple seasonal naive outperforms the Autoformer and FED-Former in Exchange and ILI (based on own experiments).
- While the main experiments are performed on multivariate datasets, the paper does not discuss how (or if) FiLM models relations between the different time series. Based on the architecture description it seems forecasts are produced independently for each channel. Authors should make this clear.
- The results on training speed presented in appendix J are not convincing. First, the total training time is more relevant than per iteration time but is not included nor discussed in the paper. Second, authors should again include simpler (non-Transformer) baselines, to better assess the trade-off between computation complexity and performance.
- Authors do not discuss the complexity or training times against the number of time series (D). As seen in Figure 5, the components operate separately for each channel, which suggests poor scaling on D. Authors should provide comparisons in datasets with more time series, such as Traffic.
- Some missing references/baselines. The N-BEATS [3] is a popular model closely related to the proposed technique, and N-HiTS [1] is an extension of the N-BEATS tailored for long-horizon forecasting.

[1] N-HiTS: Neural Hierarchical Interpolation for Time Series Forecasting, https://arxiv.org/pdf/2201.12886.pdf
[2] FreDo: Frequency Domain-based Long-Term Time Series Forecasting, https://arxiv.org/pdf/2205.12301.pdf
[3] N-BEATS: Neural basis expansion analysis for interpretable time series forecasting, https://arxiv.org/pdf/1905.10437.pdf

---

> ### Author Response · Authors · 2022-08-02
> **Response to Reviewer nNQa  [Part 1]**
>
> We would like to sincerely thank the Reviewer nNQa for providing a detailed review with insightful questions.
> >Q1: Authors should include non-Transformer methods
>
> We have run experiments with two non-transformer methods, i.e. N-HiTS and Seasonal-naive. N-HiTS is the latest development from the research group that also published N-Beats, and Seasonal-naive is a strong baseline that outperforms N-HiTs, Fedformer and Autoformer on the exchange datasets. Furthermore, we directly take the results of N-HiTS from the original paper to avoid inappropriate parameter tuning. We found that (a) FiLM outperforms N-HiTS in 33/48 cases, and (b) FiLM outperforms Seasonal-naive in all datasets, including the exchange datasets. All the updated results can be found in Appendix K.
>
> >Q2: how (or if) FiLM models relations between the different time series.
>
> Since FiLM models different channels independently, each series is indeed treated independently in this work. It is possible to mix FiLM features by introducing additional components such as MLP embedding layers. We emphasized that the main contribution of this work is to introduce the modules LPU and FEL for forecasting neural network.
>
> >Q3: The results on training speed presented in appendix J are not convincing.
>
> Thanks a lot for reviewing our paper including appendix carefully and comprehensively. We agree that the overall training time is important. In our experiments, we fix the number of training epochs, and thus comparing the training time per step is equivalent to comparing the overall training time. We are aware that the overall training time can be different if one follows a different training setting. It is true that model efficiency comparison may require different metrics for different algorithms in different scenarios.
> We acknowledged that N-HiTS is a very efficient model with less overall training time compared to FiLM. We are also aware that models, such as shallow MLP based models and seasonal naive model, can be extremely efficient for training (or even does not require any training). In the revision, we make our
> points clear by stating ``Though failing short in training speed compared to some MLP based models like N-HiTS, FiLM is an efficient model with shorter per-step training time and smaller memory usage compared to baseline models.". We finally note that the key contribution of this work is to introduce LPU and FEL modules that make long term forecasting more effective.
>
> >Q4: the complexity or training times against the number of time series (D)
>
> Yes, the training time grows linearly with dimension D. Similar to our response to Q2, for the consistency of the paper, we focus on the discussion of LPU and FEL on temporal feature and do not include any embedding layers which work on channel dimension. Actually, when D becomes too large (Exchange: D=863, Electricity: D=322), we have tested the idea of compressing channels using a linear layer and decompressing them at the output。 Indeed, a carefully compression design(low compression ratio) is necessary, but generally it works quite well in our practice.
>
> >Q5: missing references/baselines (N-BEATS and N-HiTS)
>
> Thank you for your suggestion. They are both great works, and we have cited them and compared the latest work N-HiTs in the revision (please also refer our response to Q1).
>
> >Q6: Why is the performance of FiLM different in ablation tables (table 2, 5) than from the main result table?
>
> Both Table 2 and 10 include experimental results for univariate forecasting. The results in Table 2 are intended to examine the pure effect of LPU without introducing other tricks such as multiscale mixture of experts mechanism and RevIn data normalization. This is why the results in Table 2 are worse than those in Table 10 that are obtained by the combination of all tricks. We apologize for the misleading name used for the first baseline method in Table 2, which is not our full FiLM model but an LPU+FEL combination. We will make it clear in the final version.
>
> The same argument applies to Table 5 as well, where we aim to study how different variants of LPU affect the performance and thus LPU+FEL is used as the baseline without other tricks.
>
> >Q7: Can you provide an explanation on how the model learns relations between time series or is the model univariate?
>
> As we respond to Q2, FiLM models different channels independently. We use the original features as channels. Note that an extra embedding layer could be used to merge the information between multiple time series. We are trying to apply FiLM in the recommendation domains with multiple feature embeddings. It is a topic that is worth further investigation.

---

> > ### Author Response · Authors · 2022-08-02
> > **Response to Reviewer nNQa [Part 2]**
> >
> > >Q8: How did you choose the hyperparameters (M and N) of the model for the main table. (Appendix F)
> >
> > We have performed experiments to study how M and N influence the performance on different datasets with different input lengths, and summarize the results in Figure 8 of Appendix F. Note that in the results reported in the main table, we do not use the optimal M and N as finding the optimal hyperparameters on each dataset and forecasting setting is computationally expensive. Instead, we use the default setting by fixing M=32 and N=128 for all experiments in the main table. As a general guideline from our empirical studies on different settings, we recommend either the default hyperparameter setting which can achieve decent performance (actually quite competitive), or tuning the hyperparameters via cross-validation if the optimal results is preferred given enough computing resources.

---

> > > ### Author Response · Authors · 2022-08-08
> > > **Response to Reviewer nNQa [Part 3]**
> > >
> > > Dear reviewer nNQa ,
> > >
> > > Thanks again for your review. Since the discussion period is approaching its end, we would be glad to hear from you if we have addressed your questions/concerns.
> > >
> > > Kind regards, The Authors

---

### Official Review · Reviewer_vH4a · 2022-07-12

**Rating:** 7
**Confidence:** 3
**Soundness:** 3 good
**Presentation:** 4 excellent
**Contribution:** 3 good

**Summary:**

The paper introduces two techniques to improve modeling of long time series - the Legendre Projection Unit (LPU), which compresses a time series with Legendre polynomials as basis; and the Frequency Enhanced Layer (FEL) which performs a low-rank approximation and selects a subset of Fourier transformation frequency modes. These layers are not domain or task specific and can be used in many time series modeling tasks. The authors provide both theoretical and empirical support for the effectiveness of the model.

**Questions:**

Not being from the time series community, I was a little confused by the notations at first.  It might help clarity to, for example, introduce what subscripts and superscripts mean for $f(x)_{[t-\theta, t]}$,  $ g^{(t)}(x)$, and

$ \mathbb{I}_{[t-\theta, t]}$ is before line 117.



**Strengths And Weaknesses:**

Strengths:
- Clear presentation. Figure 4 and 5 (and the code in appendix) are very informative in explaining how LPU and FEL are implemented. The extended list of ablation studies answered all of my questions about performance.
- Strong performance compared to popular deep-learning methods for long-term time series forecasting.
- Theoretically sound methods for reducing noise and capturing structure in different timescales.
- Merit in model design. One can change the basis function in LPU to other classes (Fourier / wavelets) according to data. Also love how they are modular - LPU and FEL are clear and easy to incorporate into many existing time series models to improve the problem of long horizon prediction performance deterioration. I can see wide use of methods introduced in this paper.

Weakness:
- This is not really a weakness. Both using orthogonal functions as basis to store features for time series (LPU) and using choosing Fourier frequencies / dimensional reduction to remove noise are well-explored idea in time series modeling. One may argue that the paper lacks novelty. Personally, I think it is nice to see them combined to achieve good performances and the ablation studies show the necessity of both to achieve the improvement.

---

> ### Author Response · Authors · 2022-08-02
> **Response to Reviewer vH4a**
>
> We would like to sincerely thank the Reviewer vH4a for providing valuable comments and recognizing the value of our work.
> >Q1: This is not really a weakness. Both using orthogonal functions as basis to store features for time series (LPU) and using choosing Fourier frequencies / dimensional reduction to remove noise are well-explored idea in time series modeling. One may argue that the paper lacks novelty. Personally, I think it is nice to see them combined to achieve good performances and the ablation studies show the necessity of both to achieve the improvement.
>
> Thanks for appreciating our work by acknowledging the novelty. It is worth mentioning that our designed model does not contain any non-linear activation function. All the non-linearity transformations are done using projections (Legendre function projection by LPU and Fourier base function projection by FFT). At the beginning, we even considered the title ``Projection is all you need" to emphasize the generality of our method, but later changed to the current title due to the lack of favorable results in multiple fields. It would be great if this work could inspire researchers in other fields.
>
> >Q2: Introduce what subscripts and superscripts mean before line 117.
>
> We have a smooth function $f(x)$ to approximate. f(x)_${[t-\theta,t]}$
> means $x \in [t-\theta,t]$, where $t$ is time and $\theta$ is the window
> size. $g^{(t)}(x)$ is the approximation of f(x)_${[t-\theta,t]}$. So
> $g^{(t)}(x)$ also has $x \in [t-\theta,t]$ with a window size of
> $\theta$. The $x$ in $g^{(t)}(x)$ starts from $t-\theta$ and ends at $t$
> with a window size of $\theta$. So the measure ($\mu^{(t)}$) of
> $g^{(t)}(x)$ is $\frac{1}{\theta}I_{[t-\theta,t]}(x)$. We
> will add more detailed explanations in Subsection 2.1 to clarify the
> notations.

---

### Official Review · Reviewer_Pwiu · 2022-07-13

**Rating:** 7
**Confidence:** 3
**Soundness:** 3 good
**Presentation:** 2 fair
**Contribution:** 3 good

**Summary:**

The paper introduces FiLM which stands for Frequency improved Legendre Memory Model for long-term time series forecasting. The authors leverage Legendre polynomials to obtain a fixed-sized representation of the cumulative history of the time series and combine it with Fourier analysis and low-rank approximation. The authors show that the method is competitive on long-term time series forecasting tasks and analyze the effect of several model choices and parameter sensitivities.

**Questions:**

1. The results in Table 2, Table 3, and Table 4 (LPU layer effect, low-rank approximation and mode selection policy) and on different datasets. Why is this the case? How have the datasets been selected for each experiment? I suggest that the authors provide the full table (with all datasets) of each experiment in the Appendix and clarify why these specific datasets have been selected in each experiment.

2. In Table 2 (LPU boosting results) the authors show the impact of LPUs (compared to linear layers). What does "comparable-sized linear layer" mean exactly here? Probably the biggest advantage of the LPU is to provide a meaningful representation of long time series history. However, using lagged values is a standard trick to capture long-range dependencies with standard NN architectures (like MLP layers or LSTM). I would suggest the authors include lags into this experiment to evaluate whether the LPU improves the performance over a standard architectural choice (for example, LSTM with lagged values) in forecasting.

3. In Table 2, I'm unable to link the results in the "FilM" column in either Table 1 (multivariate results) or Table 10 (univariate results). Shouldn't be the results in the FiLM correspond to either of those (at least for electricity)? The error presented in this column is also much larger than the results in Table 1/10 and the error bars in Table 9. For Electricity these results are used also in Table 5 and 6 but the results in Table 1 are different. I would kindly ask the authors to clarify this. It seems to be as expected in Tables 3 and 4.

4. Table 6: What is the relative improvement here? Relative over what? I again fail to link the results for FilM to either Table 1 or Table 10. I would kindly ask the authors to clarify the table.

**Limitations:**

The limitations of the this model or the limitations of the evaluation are not discussed in the main text. I would kindly ask the authors to add a short limitations section.

**Strengths And Weaknesses:**

The authors combine several known components (Legendre polynomials, Fourier analysis, low-rank approximation, mixture of experts) into a new time series forecasting method. While the individual components are not novel, the  non-trivial combination presented of the components presented in the paper in this paper is novel. Specifically, studying Legendre polynomials for time series forecasting could be an alternative to learning long-range dependencies in time series data. The authors provide proof for function approximation and error accumulation bounds.

The proposed model is evaluated on a set of six real-world datasets in a long-term forecasting setting and the authors carefully analyze the impact of their components in ablation experiments. However, the plethora of results presented make it hard in some cases hard on what actually has been presented and why. More detailed explanation on what exactly is presented in each experiment would improve the paper. I will detail this in the Questions section. I would increase my score if these questions are addressed during the rebuttal.

---

> ### Author Response · Authors · 2022-08-02
> **Response to Reviewer Pwiu**
>
> We would like to sincerely thank the Reviewer Pwiu for providing thorough and insightful comments.
>
> > Q1: Different datasets chosen for different ablation studies, and suggest using all datasets for ablation study
>
> Some datasets, such as Traffic and Electricity, are significantly larger than others, making it time consuming to complete its ablation study. Due to the lack of computational resources, we limit the ablation studies to the datasets of medium size. We have completed some ablation studies with all the datasets, and the rest would be finished soon. We found the same observations from the complete ablation study as those from the datasets of medium size. We will include the full results of ablation studies in the final version. Two full ablation tables are already in Appendix K.
>
> >Q2a: What does "comparable-sized linear layer" mean exactly here?
>
> The "comparable-sized linear layer" is mentioned in the ablation study of LPU. For fair comparison, the LPU layer and its variants (MLP) are set to the similar size. When putting a tensor with shape [L, D] into LPU with N Legendre functions, the output's shape becomes [N, L, D], which is N times of the input. When replacing LPU with a linear layer, it should achieve the same effect, which is [1, L, D] for input size and [N, L, D] for output size. We will make it clear in the revised version.
>
> >Q2b: Whether the LPU improves the performance over a standard architectural choice (for example, LSTM with lagged values) in forecasting?
>
> We follow the reviewer's suggesion and completed the LPU boosting experiments for LSTM with lagged values. The results are summarized in Appendix K Table 19. Lag features improved LSTM model, although not as much as LPU features. We believe this is due to the fact that LPU introduces many more global info than lag features. We also note
> that adding LPU to the lagged LSTM yields worse performance in some datasets, which could be caused by the conflicting effect between lag features and LPU features.
>
> >Q3: Unlinkable results for Table 2 and Table 10
>
> Both Table 2 and 10 include experimental results for univariate forecasting. The results in Table 2 are intended to examine the pure effect of LPU without introducing other tricks such as multiscale mixture of experts mechanism and RevIn data normalization. This is why the results in Table 2 are worse than those in Table 10 that are obtained by the combination of all tricks. We apologize for the misleading name used for the first baseline method in Table 2, which is not our full FiLM model but an LPU+FEL combination. We have made it clear in the revised  version.
>
> >Q4: Relative improvement in Table 6.
>
> Similar to Q3, the baseline method in Table 6 is LPU+FEL, not FiLM. Inspired by the review comments, we realize that we should conduct an ablation study for a full FiLM model, instead of one for LPU+FEL. We have completed the empirical studies, and included the new results in Table 6 of the revised draft.In table 6, ``relative'' means the relative increase of MSE over FiLM model. LPU+MLP combining all boosting tricks has even slightly better performance compared to FiLM for ETTm1 dataset,  but FiLM remains the most stable and effective model among all variants.

---

> > ### Author Response · Authors · 2022-08-08
> > **Response to Reviewer Pwiu [part2]**
> >
> > Dear reviewer Pwiu,
> >
> > Thanks again for your review. We just finish the experiments and update the rest two full datasets ablation tables in appendix K as you suggested. Since the discussion period is approaching its end, we would be glad to hear from you if we have addressed your questions/concerns.
> >
> > Kind regards, The Authors

---

> > ### Comment · Reviewer_Pwiu · 2022-08-09
> > **Response to authors**
> >
> > > > Q1: Different datasets chosen for different ablation studies, and suggest using all datasets for ablation study
> >
> > > Some datasets, such as Traffic and Electricity, are significantly larger than others, making it time consuming to complete its ablation study. Due to the lack of computational resources, we limit the ablation studies to the datasets of medium size. We have completed some ablation studies with all the datasets, and the rest would be finished soon. We found the same observations from the complete ablation study as those from the datasets of medium size. We will include the full results of ablation studies in the final version. Two full ablation tables are already in Appendix K.
> >
> > Thank you for providing the additional ablation results. I'm still wondering why ETTm1 is used for some ablations and ETTm2 for other ablations?
> >
> > For the shown ablations, I agree that the ablations demonstrate the gain of the LPU layer and the mode selection policy. However, the results for the low-rank approximation are mixed (error for Electricity, Traffic, and ILI increases with K=4 and K=1). Thus, I would suggest the authors to provide a balanced discussion of this result in the camera-ready version of the paper because it seems that one needs to be careful (depending on the dataset) when using LRA in this method.
> >
> > >  We follow the reviewer's suggesion and completed the LPU boosting experiments for LSTM with lagged values. The results are summarized in Appendix K Table 19. Lag features improved LSTM model, although not as much as LPU features. We believe this is due to the fact that LPU introduces many more global info than lag features. We also note that adding LPU to the lagged LSTM yields worse performance in some datasets, which could be caused by the conflicting effect between lag features and LPU features.
> >
> > Thank you for this clarification.
> >
> > > We follow the reviewer's suggesion and completed the LPU boosting experiments for LSTM with lagged values. The results are summarized in Appendix K Table 19. Lag features improved LSTM model, although not as much as LPU features. We believe this is due to the fact that LPU introduces many more global info than lag features. We also note that adding LPU to the lagged LSTM yields worse performance in some datasets, which could be caused by the conflicting effect between lag features and LPU features.
> >
> > Thank you for adding this experiment. Table 19 suggests that the used lag features do NOT improve the LSTM model (values in the "LPU" column are higher for lagged-LSTM than for LSTM). What lags are exactly used? My expectation would have been that the lagged LSTM should be better than the vanilla LSTM. I suggest to add these details in the final version of the paper.
> >
> > > Both Table 2 and 10 include experimental results for univariate forecasting. The results in Table 2 are intended to examine the pure effect of LPU without introducing other tricks such as multiscale mixture of experts mechanism and RevIn data normalization. This is why the results in Table 2 are worse than those in Table 10 that are obtained by the combination of all tricks. We apologize for the misleading name used for the first baseline method in Table 2, which is not our full FiLM model but an LPU+FEL combination. We have made it clear in the revised version.
> >
> > > Similar to Q3, the baseline method in Table 6 is LPU+FEL, not FiLM. Inspired by the review comments, we realize that we should conduct an ablation study for a full FiLM model, instead of one for LPU+FEL. We have completed the empirical studies, and included the new results in Table 6 of the revised draft.In table 6, ``relative'' means the relative increase of MSE over FiLM model. LPU+MLP combining all boosting tricks has even slightly better performance compared to FiLM for ETTm1 dataset, but FiLM remains the most stable and effective model among all variants.
> >
> > Thank you for this clarification. I find this still somewhat confusing in the paper (it is not clear to me at which places this is corrected in the current version). I also noted that several of the added Appendix tables/captions contain typos. I think the paper still needs significant edits for the camera-ready version.
> >
> > Overall, I think the proposed method is an interesting addition to the time series forecasting landscape, but I remain skeptical of the effectiveness of the low-rank approximation. The authors addressed most of my concerns and I raised my score.

---

> > > ### Author Response · Authors · 2022-08-09
> > > **Response to Reviewer Pwiu [part3]**
> > >
> > > Dear Reviewer Pwiu
> > >
> > > We want to thank your valuable comments sincerely. Indeed as you point out, the low-rank approximation is not a stable improvement design for all datasets. On the contrary, it will hurt our performance in the heaviest compression version. But, it might be used as a building block for a future efficient or mobile model as it dramatically decreases the learnable parameters. And thanks for pointing out the typos; we will check the appendix and fix them.
> > >
> > > Kind regards, The Authors

---

### Meta-Review · Area_Chair_2Kz3 · 2022-08-27

**Recommendation:** Accept
**Confidence:** Certain

**Metareview:**

Paper provides a time series modeling technique combining the use of Legendre polynomials for projections and Frequency based low rank approximation / selection.
The reviewers found the paper to be interesting, and the results convincing and possibly usable in other sequence modeling tasks.
Some questions were raised by nNQa  about the baselines / comparisons, that I felt were addressed appropriately by the authors. Other questions that were raised about the details of the experiments, including the datasets, the ablations performed, and comparisons to alternatives (such as lagged inputs in LSTMs, and comparisons to n-Hits) seem to have been well addressed by the authors.


**Award:**

No

---

### Decision · Program_Chairs · 2022-09-14

Accept